# Glucocorticoid receptor-PPARα axis in fetal mouse liver prepares neonates for milk lipid catabolism

Gianpaolo Rando[1†], Chek Kun Tan[2†], Nourhène Khaled[1], Alexandra Montagner[3], Nicolas Leuenberger[1], Justine Bertrand-Michel[4], Eeswari Paramalingam[2], Hervé Guillou[3], Walter Wahli[1,2,3*]

[1]Center for Integrative Genomics, University of Lausanne, Lausanne, Switzerland; [2]Lee Kong Chian School of Medicine, Nanyang Technological University, Singapore; [3]UMR 1331 ToxAlim Research Centre in Food Toxicology, INRA, Université de Toulouse, Toulouse, France; [4]IFR 150 Plateforme Metatoul, Institut Fédératif de Recherche Bio-Médicale de Toulouse INSERM U563, Toulouse, France

**Abstract** In mammals, hepatic lipid catabolism is essential for the newborns to efficiently use milk fat as an energy source. However, it is unclear how this critical trait is acquired and regulated. We demonstrate that under the control of PPARα, the genes required for lipid catabolism are transcribed before birth so that the neonatal liver has a prompt capacity to extract energy from milk upon suckling. The mechanism involves a fetal glucocorticoid receptor (GR)-PPARα axis in which GR directly regulates the transcriptional activation of PPARα by binding to its promoter. Certain PPARα target genes such as *Fgf21* remain repressed in the fetal liver and become PPARα responsive after birth following an epigenetic switch triggered by β-hydroxybutyrate-mediated inhibition of HDAC3. This study identifies an endocrine developmental axis in which fetal GR primes the activity of PPARα in anticipation of the sudden shifts in postnatal nutrient source and metabolic demands.

*For correspondence: walter. wahli@ntu.edu.sg

†These authors contributed equally to this work

Competing interests: The authors declare that no competing interests exist.

## Introduction

In mammals, embryonic and postnatal development depends on nutrition from placentation and lactation, respectively (*Brawand et al., 2008*).Before birth, hepatic energy metabolism relies mainly on glucose catabolism. Metabolic fluxes change abruptly at birth when milk, which has a higher lipid but relatively lower glucose content, becomes the exclusive nutrient (*Girard et al., 1992*). At the first few hours after birth, liver expresses the rate-limiting enzymes responsible for extracting energy from milk (*Krahling et al., 1979*; *Huyghe et al., 2001*). However, whether lipid catabolism at birth is developmentally programmed or an adaptive response requiring an external stimulus remains unknown. Failure to adapt to this catabolic switch results in life-threatening errors of metabolism, with serious energy imbalances that are recapitulated in mouse models of neonatal liver steatosis (*Ibdah et al., 2001*; *Cherkaoui-Malki et al., 2012*).

Peroxisome proliferator-activated receptor α (PPARα) is a key transcriptional regulator of lipid metabolism owing to its activation by a lipid surge to induce lipid catabolism (*Desvergne et al., 2006*; *Montagner et al., 2011*, *2016*). However,the role of PPARα in the perinatal liver is not fully understood. Certain PPARα target genes (e.g., acyl-CoA oxidase 1 [*Acox1*]) are highly expressed in fetal liver at the end of gestation, whereas the expression of others (e.g., fibroblast growth factor 21 [*Fgf21*]) strictly depends on postnatal milk uptake (*Hondares et al., 2010*; *Yubero et al., 2004*). Based on these contrasting observations, we hypothesized the presence of a PPARα-triggering

**eLife digest** Birth is a highly stressful and critical event. In the womb, babies rely on the supply of oxygen and nutrients provided by the placenta. However, once they are born they need to breathe for themselves and gain all their nutrients from suckling milk. The placenta provides a sugar-rich diet, while milk is richer in fat. Failing to cope with this change in diet leads to serious complications and sometimes death. Therefore, a better understanding of how the body adapts to these changes may shed light on pathways that are important for good health in later life.

The liver plays a central role in processing the nutrients absorbed by the gut. It uses fats to produce molecules called ketone bodies, such as β-hydroxybutyrate, which are then used as fuel by other tissues and organs including the heart, muscle and the brain. A protein called PPARα controls the production of ketone bodies primarily by regulating genes that are involved in the uptake and breakdown of fat in the liver. However, little is known about how this protein affects the development of the liver.

Here, Rando, Tan et al. report that mice start to produce more PPARα in the liver shortly before birth. This ultimately activates several genes that encode enzymes that break down fats. The experiments show that during labor, stress hormones called glucocorticoids directly stimulate the production of PPARα in the liver of the fetus to prepare newborn mice for harnessing energy from fat-rich milk.

In the absence of PPARα, mouse liver cells are less able to break down fats after birth and so start to accumulate fat, resulting in fewer ketone bodies being produced. Rando, Tan et al. show that β-hydroxybutyrate regulates some PPARα target genes, including one called *Fgf21*. The activity of this gene increases only after milk suckling starts and it encodes a protein that enhances the breakdown of fats in the liver. Without PPARα, the expression levels of its target genes, including *Fgf21*, do not increase after birth, which promotes the build up of fats in liver cells, a condition known as liver steatosis.

Overall, the results reported by Rando, Tan et al. highlight how stress during labor plays an important role in priming the body to cope with a fat-rich diet after birth. Future studies will need to determine if stress hormones and ketone bodies could be used as therapies for babies born by caesarean section with liver steatosis.

program before birth that does not require milk suckling for activation. Given that fetal liver metabolism mostly relies on glycolysis rather than lipid oxidation, we postulated that the liver phenotype of PPARα-deficient fetuses may not be obvious due to the absence of a lipid-rich diet challenge.

By measuring gene expression in prenatal and postnatal livers, we show that PPARα activation has already occurred a few days before birth, and that glucocorticoid receptor (GR)-mediated control of PPARα in late gestation prepares its physiological role in the pups for harnessing milk lipids immediately after birth. However, before birth we could barely detect the expression of certain PPARα target genes such as *Fgf21*, which is exclusively involved in adaptive metabolism. Therefore, we explored the mechanistic basis for the temporal regulation of PPARα target genes. We found that certain PPARα target genes, including *Fgf21* and *Angptl4* (Angiopoietin-like 4), are epigenetically controlled by histone deacetylase 3 (HDAC3) and de-repressed in response to β-hydroxybutyrate, a by-product of fatty acid oxidation (FAO). Taken together, our data provide the evidence of a major role of glucocorticoid signaling in direct hepatic regulation of PPARα and indirect HDAC3-mediated regulation of FGF21, which controls important metabolic and thermogenic events in the early days of life (*Hondares et al., 2010*).

## Results

### GR controls PPARα expression in the late fetus

Stress at labor is associated with high glucocorticoid signaling (*Barlow et al., 1974*). We previously reported that glucocorticoids stimulate PPARα expression in the adult liver, but the mechanism was not elucidated (*Lemberger et al., 1994*). Interestingly, the mRNA expression of GR (*Nr3c1*) in the

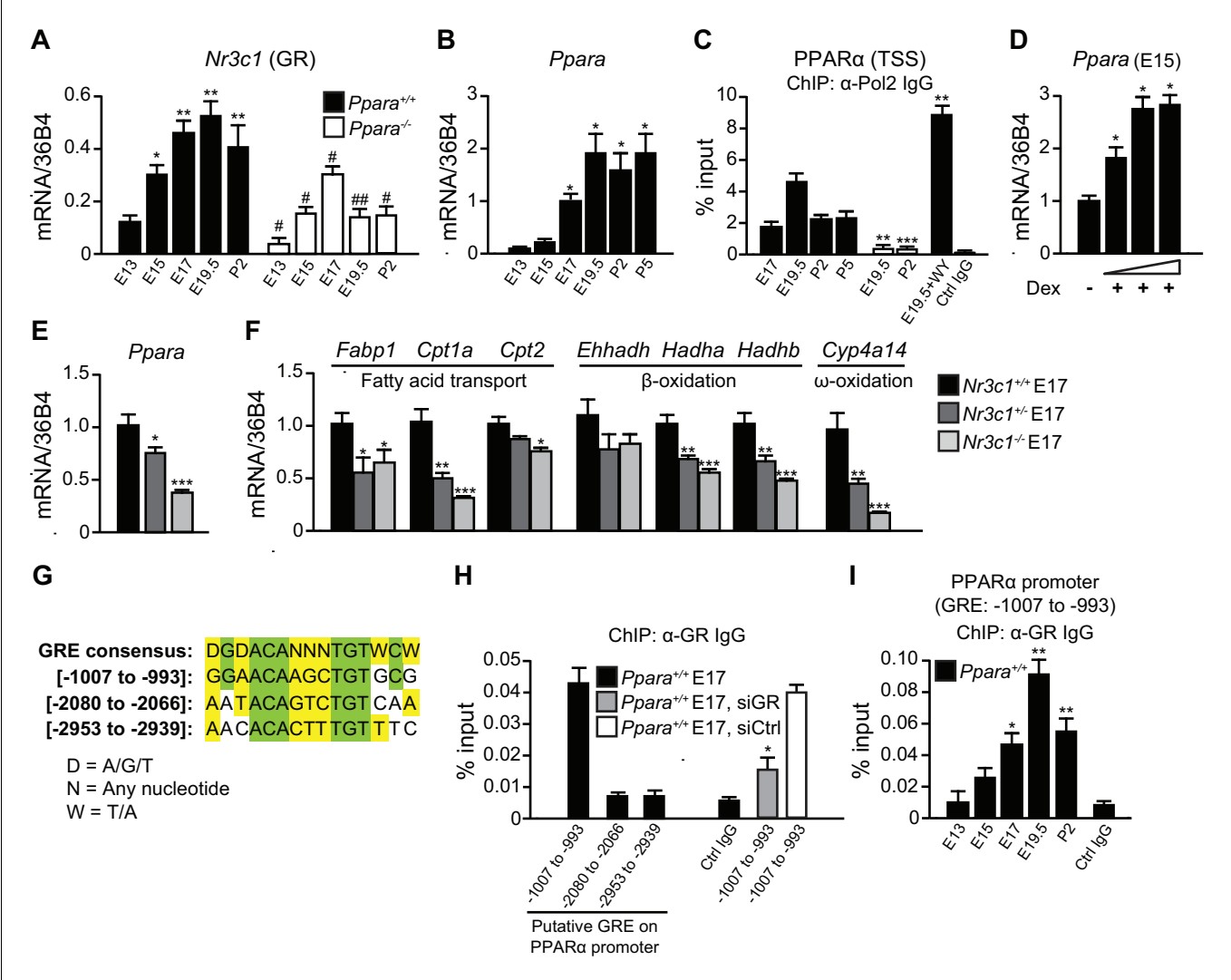

**Figure 1.** GR directly controls fetal PPARα expression. (A, B) Ontogenic expression of *Nr3c1* (A) and *Ppara* (B) mRNA in the developing mouse liver. *p<0.05, **p<0.01 vs. E13 samples; #p<0.05, ##p<0.01 vs. WT counterparts. (C) Enrichment of the DNA fragments containing the PPARα TSS using anti-Pol2 antibodies or pre-immune control IgG with or without PPARα agonist (WY-14643) treatment in pregnant dams. **p<0.01, ***p<0.001 vs. respective WT counterparts without WY-14643 treatment. (D) *Ppara* mRNA levels in the E15 *Ppara*$^{+/+}$ liver explants with or without dexamethasone (Dex) treatment for 24 hr. Dex concentrations of 0.1 μM, 1 μM, and 10 μM were used. The vehicle control group was treated with 0.1% ethanol. *p<0.05 vs. vehicle control. (E,F) mRNA expression of PPARα and its target genes in the fetal livers of GR models at E17. *p<0.05, ** p<0.01, ***p<0.001 vs. *Nr3c1*$^{+/+}$ liver. (G) Alignment of the GRE consensus sequence with the three putative GRE sequences located upstream of the PPARα promoter. (H) Enrichment of the DNA fragment containing the three putative GRE found within the PPARα promoter at regions spanning −1007 to −993, −2080 to −2066, and −2953 to −2939 in fetal liver at E17 using anti-GR antibody or pre-immune control IgG. Enrichment levels were expressed as the percentage input. GR-targeting siRNA was used to knockdown *Nr3c1* expression and to determine the specificity of GR binding to this putative GRE. *p<0.05 vs. non-targeting siRNA treatment group. (I) Enrichment of GRE spanning -1007 to −993 of the PPARα promoter in *Ppara*$^{+/+}$ liver during development. *p<0.05, **p<0.01 vs. E13 samples. Data are presented as mean ± SEM; n = 4–6. Statistical analyses were performed using two-tailed Mann-Whitney tests.

fetal liver is the highest just before birth (*Speirs et al., 2004*).Because GR plays a role in the preparation for birth (*Cole et al., 1995*), we further tested GR implication in the regulation of fetal PPARα expression. We confirmed that *Nr3c1* mRNA levels in the fetal liver peak at embryonic day E19.5, similar to *Ppara* mRNA expression (*Figure 1A,B*). Notably, *Ppara* mRNA levels were low in the liver at E13 and E15, but markedly increased at E17, peaking just before birth at E19.5 (*Figure 1B*). This observation coincides with a maximal RNA polymerase 2 (Pol2) recruitment to the PPARα transcriptional start site (TSS) (*Figure 1C*) and enhanced nuclear accumulation of PPARα protein similar to

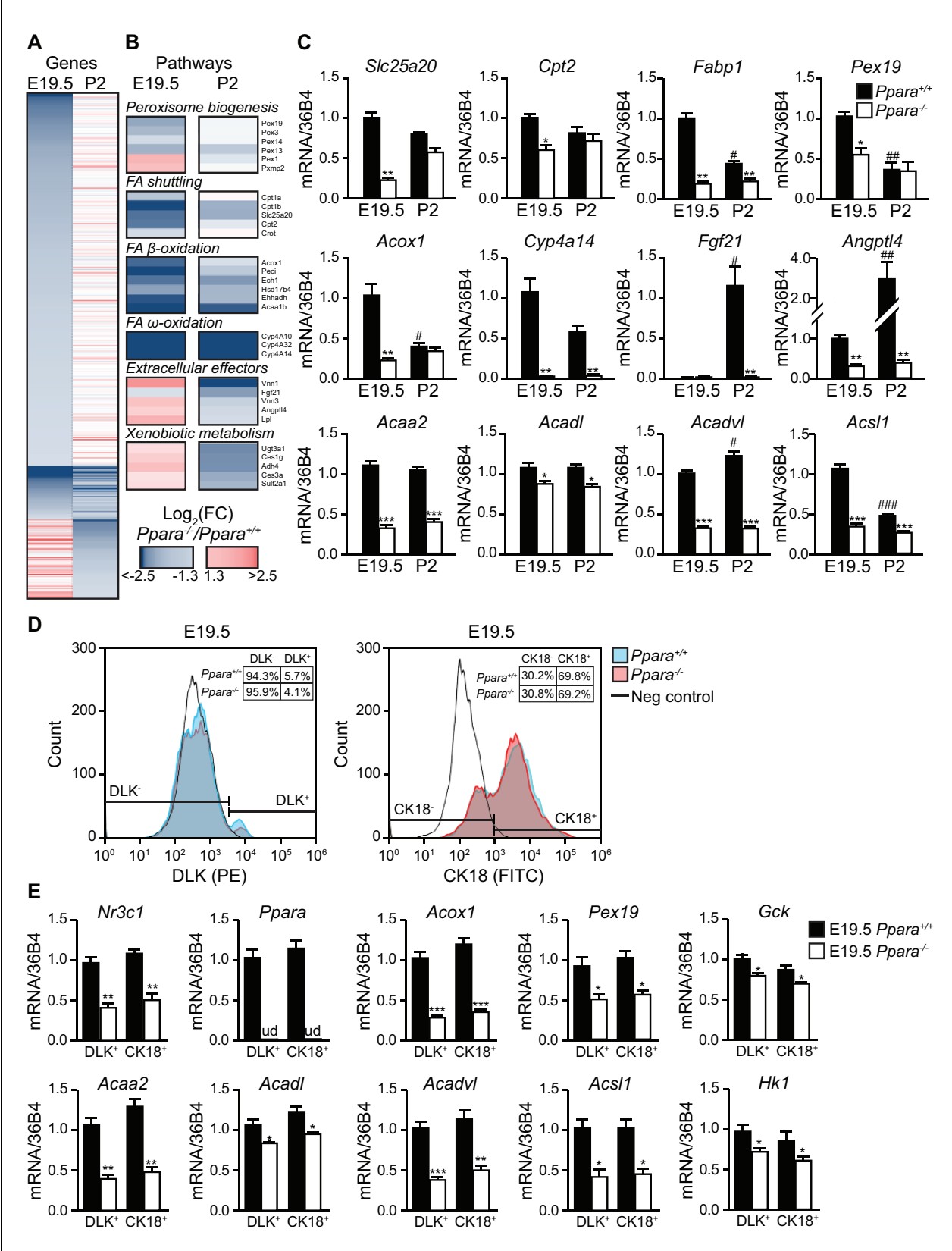

**Figure 2.** PPARα is a functional transcription factor in the term fetus. (**A**) Heat map of genes significantly altered at E19.5 (left panel) or P2 (right panel) in *Ppara⁻/⁻* livers (red: up-regulation; blue: down-regulation). Genes were grouped by fetal (E19.5) or postnatal (P2) expression and ordered by strength

*Figure 2 continued on next page*

*Figure 2 continued*

of regulation based on the logarithmic fold change (log$_2$FC). (**B**) Gene ontology summarizing prenatal and postnatal PPARα-mediated regulation of metabolic pathways. (**C**) mRNA levels of representative PPARα target genes with differential fetal (E19.5) and postnatal (P2) regulation in *Ppara*$^{-/-}$ and *Ppara*$^{+/+}$ livers. *Slc25a20*: carnitine translocase; *Cpt2*: carnitine palmitoyltransferase 2; *Fabp1*: fatty acid binding protein 1; *Pex19*: peroxisome biogenesis factor 19; *Acox1*: peroxisomal acyl-CoA oxidase 1; *Cyp4a14*: cytochrome P450 4A14; *Fgf21*: fibroblast growth factor 21; *Angptl4*: angiopoietin-like protein 4; *Acaa2*: acetyl-CoA acyltransferase 2; *Acadl*: acyl-CoA dehydrogenase, long chain; *Acadvl*: acyl-CoA dehydrogenase, very long chain; *Acsl1*: acyl-CoA synthetase, long-chain family member 1. Data are presented as mean ± SEM; n = 6, *p<0.05, **p<0.01, ***p<0.001 vs. wild-type controls; #p<0.05, ##p<0.01, ###p<0.001 vs. respective E19.5 samples. (**D**) Flow cytometric analyses of hepatoblasts (DLK$^+$) and hepatocytes (CK18$^+$) in fetal livers at E19.5. (**E**) mRNA expression levels of *Nr3c1*, *Ppara*, and PPARα target genes involved in peroxisome biogenesis (*Pex19*), peroxisomal and mitochondrial FAO (*Acox1*, *Acaa2*, *Acadl*, *Acadvl*, and *Acsl1*) in sorted fetal hepatoblast and hepatocyte fractions. Glycolytic genes (e.g., *Gck* and *Hk1*) were also investigated in parallel with oxidative genes. *Nr3c1*: glucocorticoid receptor; *Ppara*: peroxisome proliferator-activated receptor α; ud: undetermined; *Gck*: glucokinase; *Hk1*: hexokinase 1. Data are presented as mean ± SEM; n = 4, *p <0.05, **p<0.01, ***p<0.001 vs. wild-type controls.

The following source data and figure supplement are available for figure 2:

**Source data 1.** A list of PPARα-regulated genes and pathways in prenatal and neonatal livers.

**Figure supplement 1.** PPARα deficiency leads to compensatory up-regulation of genes.

the levels in the postnatal pups (Figure 5A). Interestingly, we also observed a significant reduction in the expression of *Nr3c1* in *Ppara*$^{-/-}$ livers during development when compared with *Ppara*$^{+/+}$ controls (*Figure 1A*), suggesting that PPARα may also reciprocally regulate *Nr3c1* expression. To investigate the regulation of PPARα expression by GR, we treated E15 fetal liver explants, when *Ppara* expression is relatively low, with GR agonist dexamethasone, which induced *Ppara* mRNA expression in a dose-dependent manner (*Figure 1D*). We also examined the expression of *Ppara* in GR-null fetuses at E17 (*Figure 1E*), when *Ppara* mRNA levels just begin to increase in the wild-type fetal liver (*Figure 1B*). GR exerted a gene dosage effect on the expression of both *Ppara* and its target genes, including *Cpt1a*, *Hadha*, *Hadhb*, and *Cyp4a14* (*Figure 1E,F*).

Next, we addressed the mechanisms involved in priming the high expression and activity of PPARα just before birth. We identified three putative GR response elements (GRE) in the promoter/regulatory region of *Ppara* and analyzed their occupancy by GR in the fetal liver at E17 (*Figure 1G*). Chromatin immunoprecipitation (ChIP) using an antibody against GR revealed that the element spanning -1007 to -993 was preferentially occupied while the other two sites were not (*Figure 1H*). Upon treatment of primary hepatocytes with GR-targeting siRNA, the occupancy of GR at this putative site was significantly decreased as compared to treatment with non-targeting siRNA, thereby indicating the specificity of GR binding to this site (*Figure 1H*). Binding of GR to this putative GRE also followed the ontogenic pattern of *Ppara* expression in the fetal liver (Compare *Figure 1I* with *Figure 1B*). Therefore, PPARα is a direct target of GR in the fetal liver and its expression parallels that of GR in later stages of fetal development. These findings reveal a novel endocrine GR-PPARα axis in the regulation of the fatty acid (FA) catabolic machinery just before birth.

## PPARα controls the prenatal lipid catabolic machinery

We next performed microarray analysis on gene expression in the fetal and neonatal *Ppara*$^{+/+}$ and *Ppara*$^{-/-}$ livers. PPARα-dependent fold changes in gene expression were higher in the term fetuses than in suckling pups, revealing major transcriptional PPARα activities just before delivery (*Figure 2A*; *Figure 2— figure supplement 1A,B*). We found 915 and 425 down-regulated genes in *Ppara*$^{-/-}$ liver at E19.5 and postnatal day 2 (P2), respectively (*Figure 2A*). Many of the differentially expressed genes were preferentially regulated in either the prenatal or postnatal liver, suggesting the involvement of distinct metabolic pathways at these two time points (*Figure 2A,B*; *Figure 2— figure supplement 1B*; *Figure 2—source data 1*, SD12). Genes preferentially controlled by PPARα in fetal liver (e.g., *Pex19*, *Slc25a20*, *Cpt2*, and *Fabp1*) relate to peroxisome biogenesis and FA shuttling−two upstream steps essential for FA catabolism (*Figure 2B,C*). Genes encoding the peroxisomal and mitochondrial enzymatic core of β- and ω-FAO, such as *Acox1*, *Acaa2*, *Acadl*, *Acadvl*, *Acsl1*, and *Cyp4a14*, were prenatally and/or postnatally controlled by PPARα (*Figure 2B,C*). In contrast, genes encoding the liver-secreted adaptive effectors of lipid catabolism (e.g., *Fgf21*) and xenobiotic-detoxifying enzymes (e.g., *Ugt3a1*, *Ces1g*, *Adh4*, *Ces3a*, *Sult2a1*) were only stimulated by

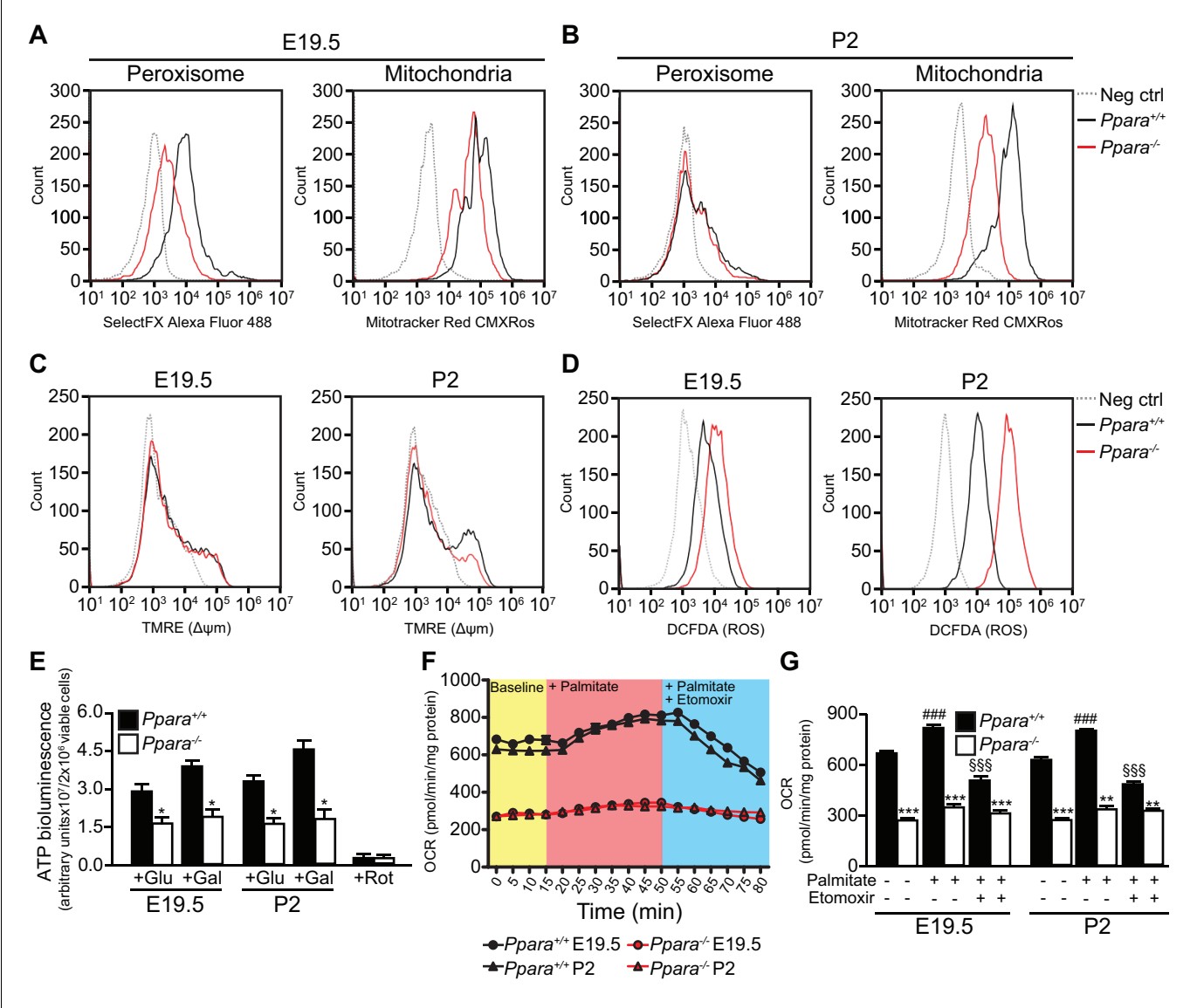

**Figure 3.** Defective mitochondrial function, fatty acid oxidation and energy production in *Ppara*⁻/⁻ hepatocytes. (A,B) Flow cytometric analyses of intracellular peroxisomes and mitochondria in primary hepatocytes isolated from *Ppara*⁻/⁻ and *Ppara*⁺/⁺ livers at E19.5 (A) and P2 (B) using Alexa Fluor 488-labeled antibodies against peroxisome membrane protein 70 and Mitotracker Red, respectively. (C,D) Flow cytometric analyses of mitochondrial membrane potential ($\Delta\psi_M$) (C) and intracellular ROS (D) in primary hepatocytes isolated from *Ppara*⁻/⁻ and *Ppara*⁺/⁺ livers at E19.5 and P2 using tetramethylrhodamine, ethyl ester (TMRE) and 2',7'-dichlorofluorescin diacetate (DCFDA), respectively. (E) ATP production in primary hepatocytes isolated from *Ppara*⁻/⁻ and *Ppara*⁺/⁺ livers (n = 4 per group) in the presence of glucose (+Glu), galactose (+Gal), or rotenone (+Rot, a mitochondrial electron transport chain complex I inhibitor) as a negative control. Values represent arbitrary bioluminescence units normalized to the number of viable cells. (F,G) Oxygen consumption rates (OCRs) in primary hepatocytes isolated from *Ppara*⁻/⁻ and *Ppara*⁺/⁺ livers in the presence or absence of palmitate and etomoxir, a mitochondrial β-oxidation inhibitor. Data are presented in time-lapse (F) and treatment end points (G) at 15 min for basal respiration, 50 min for palmitate treatment, and 80 min for palmitate cum etomoxir treatment. Data represent mean ± SEM; n = 3–9 unless otherwise stated, **p<0.01, ***p<0.001 vs. wild-type controls; ###p<0.001 vs. no treatment group; §§§p<0.001 vs. palmitate treatment group (two-tailed Mann-Whitney test).

PPARα postnatally (*Figure 2B,C*). Focused gene expression profiling further confirmed that genes encoding the rate-limiting enzymes involved in FA shuttling, mitochondrial and peroxisomal FA β-oxidation, and microsomal FA ω-oxidation were concomitantly down-regulated in *Ppara*⁻/⁻ liver before and/or after birth (*Figure 2C*). Lack of PPARα also resulted in fetal liver in an up-regulation of

other nuclear receptors and their target genes involved in lipid metabolism (*Figure 2—figure supplement 1C,D*).

Since there is a heterogeneous population of undifferentiated hepatoblasts and differentiated hepatocytes in the liver at E19.5, we next determined whether one of these two cell populations was predominantly affected by the PPARα-dependent changes in gene expression. Using antibodies against hepatoblast- and hepatocyte-specific markers (i.e., Delta-Like Homolog 1 (DLK) and cytokeratin 18 (CK18), respectively), we performed flow cytometric analyses on the cells extracted from *Ppara*$^{+/+}$ and *Ppara*$^{-/-}$ livers at E19.5. Upon cell sorting, we specifically recovered the DLK$^+$ and CK18$^+$ fractions. Firstly, we showed that PPARα deficiency only led to marginal changes in these two hepatic cell populations (*Figure 2D*). Secondly, we determined the mRNA expression of *Nr3c1*, *Ppara* and the target genes of PPARα in these cells. Our results indicated that both *Ppara*$^{+/+}$ hepatoblasts and hepatocytes similarly contributed to the expression of these genes (*Figure 2E*). Notably, we observed a modest but not significant upward trend in the expression of oxidative genes (e.g., *Acox1*, *Acaa2*, *Acadl*, and *Acadvl*) and a downward trend in the expression of glycolytic genes (e.g., *Gck* and *Hk1*) in the *Ppara*$^{+/+}$ hepatocytes compared to hepatoblasts, possibly suggesting a more pronounced oxidative program in hepatocytes (*Figure 2E*). In contrast, the expression of these genes was concomitantly down-regulated in *Ppara*$^{-/-}$ hepatoblasts and hepatocytes, thereby highlighting the pivotal role of PPARα in glycolytic and oxidative metabolism in both cell types.

Consistent with our gene expression studies, *Ppara*$^{-/-}$ fetal livers had reduced numbers of peroxisomes and mitochondria compared to wild-type littermates at E19.5 (*Figure 3A*). However, only the mitochondria numbers but not the peroxisome numbers varied between the two groups at P2 (*Figure 3B*), underscoring the crucial role of mitochondrial FAO in PPARα-mediated lipid catabolism after birth. We next measured mitochondrial membrane potential ($\Delta_{\Psi m}$) and intracellular reactive oxygen species (ROS) production levels which are two common parameters used in assessing mitochondria health and functions (*Suski et al., 2012*). Interestingly, we observed a much lower $\Delta_{\Psi m}$ and a markedly increased intracellular ROS level in *Ppara*$^{-/-}$ primary hepatocytes at P2 (*Figure 3C,D*), indicative of mitochondria dysfunction at this stage. However, there was no discernible difference in $\Delta_{\Psi m}$ between *Ppara*$^{+/+}$ and *Ppara*$^{-/-}$ hepatocytes at E19.5 despite that a slight increase in the intracellular ROS level was observed in the latter. These findings suggest that an increased ROS production in cells likely precedes the loss of $\Delta_{\Psi m}$ and mitochondria dysfunction, consistent with a previous report of a similar mitochondrial disorder associated with the dysfunctions of the respiratory chain components (*Lebiedzinska et al., 2010*). Indeed, respiratory chain-mediated ATP production in *Ppara*$^{-/-}$ hepatocytes was approximately 35–50% of the levels measured in both prenatal and postnatal wild-type livers (*Figure 3C*). Substantiating these findings, a reduction of ~2.3-fold was observed in the basal oxygen consumption rate in *Ppara*$^{-/-}$ primary hepatocytes from both E19 and P2 livers (*Figure 3D,E*). In the presence of FA (i.e., palmitate), the oxygen consumption rate significantly increased in *Ppara*$^{+/+}$ hepatocytes but remained very similar in *Ppara*$^{-/-}$ hepatocytes, indicating a deficiency in the activation of FAO in the latter (*Figure 3D,E*). The addition of etomoxir, an inhibitor of carnitine palmitoyltransferase 1, resulted in significant inhibition of mitochondrial β-FAO in *Ppara*$^{+/+}$ but not in *Ppara*$^{-/-}$ hepatocytes. *Ppara*$^{+/+}$ hepatocytes exhibited a robust mitochondrial respiratory capacity of ~40% at both time points (*Figure 3E*). In contrast, only ~25% and ~11% of the total oxygen consumption rate were attributed to mitochondrial β-FAO in *Ppara*$^{-/-}$ hepatocytes from E19 and P2 livers, respectively. Thus, these results provide evidence that PPARα-dependent respiration occurs in the term fetus. The higher expression of FA catabolic genes in fetal liver just before birth may prime the liver for the upcoming postnatal energy demand or point to an unknown role of fetal FAO. Therefore, we investigated whether the above defects in *Ppara*$^{-/-}$ hepatocytes contribute to any phenotype in the fetal and neonatal liver.

## PPARα is required for protection against neonatal hepatic steatosis

Gross examination and histological analysis of the fetal *Ppara*$^{-/-}$ livers at E19.5 did not reveal any discernible differences when compared with *Ppara*$^{+/+}$ controls. However, after birth, steatosis developed in *Ppara*$^{-/-}$ livers at P2 but not in *Ppara*$^{+/+}$ livers (*Figure 4A,B*). Compared to *Ppara*$^{+/+}$ littermates, *Ppara*$^{-/-}$ pups displayed an enlarged and pallid liver with neutral lipid accumulation demonstrated by Oil Red O staining (*Figure 4A,B*). *Ppara*$^{-/-}$ livers contained significantly higher levels of triglycerides and cholesterol ester at P2 with a concomitant increase in blood triglycerides (*Figure 4D–F*). In contrast, *Ppara*$^{-/-}$ pups also exhibited markedly reduced serum β-hydroxybutyrate

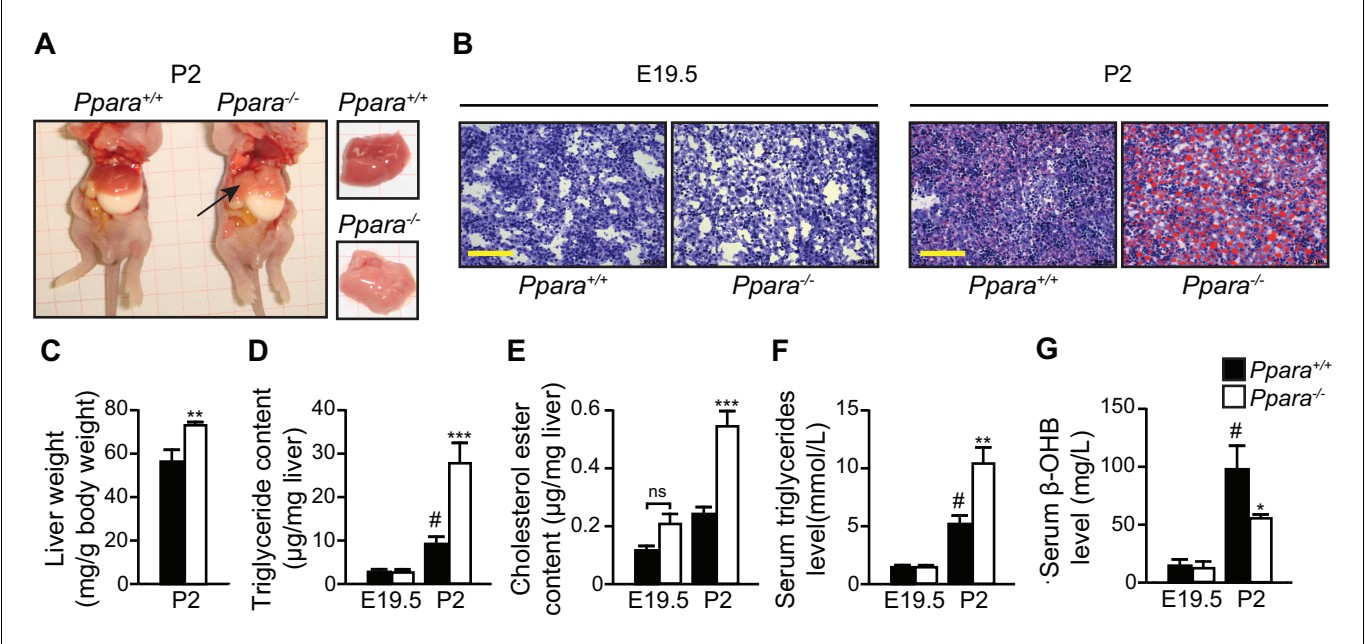

**Figure 4.** PPARα deficiency leads to congenital hepatic steatosis after birth. (**A**) Photographs showing dissected *Ppara*-/- and *Ppara*+/+ pups (*left*) and livers (*right*). The pallid liver of a *Ppara*-/- pup is indicated by an arrow. The white content of the stomach indicates milk ingestion. (**B**) Representative Oil Red O-stained liver sections counterstained with methylene blue. Scale bars = 100 μm. (**C–G**) The Mean body weight (**C**), triglyceride (**D**) and cholesterol ester (**E**) contents of the liver, and serum levels of triglyceride (**F**) and β-hydroxybutyrate (β-OHB) (**G**). Data are presented as mean ± SEM; n = 6, *$p<0.05$, **$p<0.01$, ***$p<0.001$ vs. wild-type controls; #$p<0.05$ vs. respective E19.5; ns, not significant (two-way ANOVA with Bonferroni post-hoc analysis).

The following figure supplements are available for figure 4:

**Figure supplement 1.** Postnatal lipid catabolic derangements in *Ppara*-/- pups.

**Figure supplement 2.** Anaplerotic compensation for defective oxidative metabolism and ketogenesis in suckling *Ppara*-/- pups.

levels (i.e., hypoketonemia) and impaired essential FA profiles (*Figure 4G*, *Figure 4—figure supplement 1A,B*). All of these anomalies were absent in *Ppara*-/- fetuses at E19.5. Interestingly, the postnatal phenotype of liver steatosis observed in *Ppara*-/- pups spontaneously and gradually resolved at P15, coinciding with the suckling-to-weaning transition period when carbohydrate-rich food gradually replaces the lipid-rich milk (*Figure 4—figure supplement 1C,D*). These findings suggest that the neonatal steatosis observed in *Ppara*-/- pups may be attributed to the ingestion of milk lipids.

Pups begin to nibble solid food in their second week, before weaning. To test whether the change in food composition that occurs at weaning is responsible for the reduction in the fatty liver phenotype as observed in *Ppara*-/-pups after P5-6, we performed a 10-day high-fat, low-carbohydrate diet challenge starting just before P5 and continuing into the suckling-to-weaning transition period. After the high-fat diet (HFD) challenge, the occurrence of liver steatosis was reenacted at P15 in *Ppara*-/- mice which consistently exhibited significantly higher liver weight, hepatic and serum triglyceride levels (*Figure 4—figure supplement 1E–H*). In contrast, liver steatosis was absent in the *Ppara*+/+ counterparts fed the same diet as well as in mice fed the control diet (*Figure 4—figure supplement 1E– H*). Furthermore, we also demonstrated in hepatocyte-specific *Ppara*-/- mice (i.e., *Ppara*fl/fl*Alb*Cre/+) the presence of liver steatosis at P3, thereby confirming that the phenotypes observed in *Ppara*-/- mice is most likely due to the hepatocyte-specific effects (*Figure 4— figure supplement 1I*) (*Montagner et al., 2016*). Taken together, PPARα deficiency leads to lower mitochondria numbers, defective FA oxidative metabolism, and aberrant mitochondria functions which collectively contribute to the development of liver steatosis and hypoketonemia in *Ppara*-/- neonates as a result of their inability to metabolize milk lipids.

An effective hepatic FAO is necessary to support hepatic de novo ketogenesis and gluconeogenesis by providing essential co-factors such as acetyl-CoA and/or NADH (*Girard, 1986*). This notion is consistent with the presence of hypoketonemia in *Ppara*[-/-] neonates with defective hepatic FAO. Interestingly, we found that *Ppara*[-/-] pups were able to maintain a body weight increase similar to their *Ppara*[+/+] littermates (*Figure 4—figure supplement 2A*), suggesting that *Ppara*[-/-] neonates harnessed energy required for normal growth from fuels other than ketone bodies and lipids. Furthermore, we did not detect any differences in the plasma glucose levels between the two genotypes during the suckling period from P2 to P15 (*Figure 4—figure supplement 2B*). This was rather surprising since *Ppara*[-/-] neonates were previously reported to exhibit hypoglycemia at P1 due to impaired gluconeogenesis from glycerol (*Cotter et al., 2014*). This finding prompted us to check whether the gluconeogenesis pathway is modified in the *Ppara*[-/-] liver at P2. Based on our microarray data, we found that most of the genes (e.g., *Gpt*, *Pck1*, and *G6pc*) encoding the rate-liming enzymes

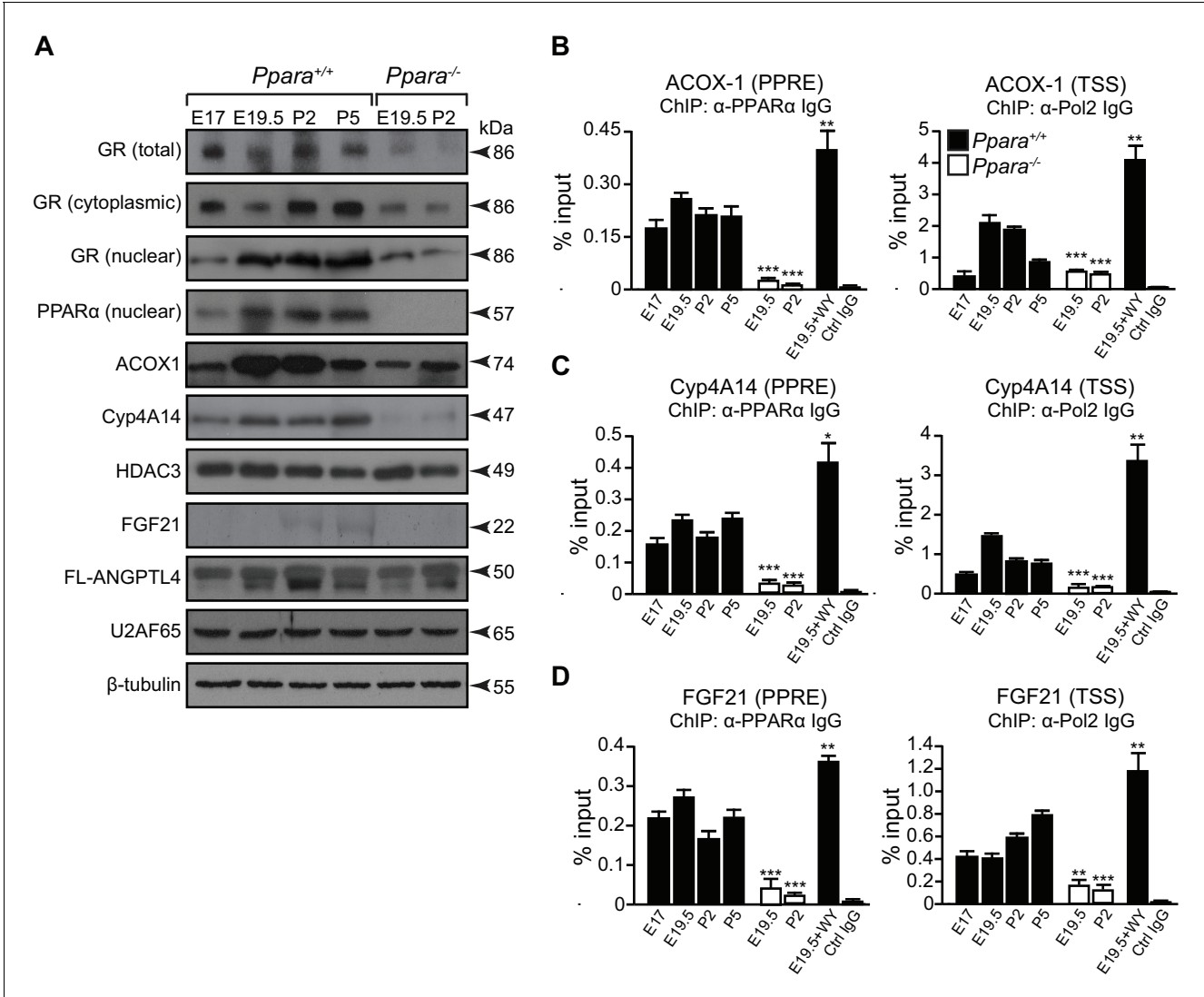

**Figure 5.** A temporal dichotomy in the regulation of PPARα target genes before and after birth. (**A**) Immunoblots showing the ontogenic expression of cytoplasmic and nuclear GR, HDAC3, PPARα, and its target genes, including ACOX1, Cyp4A14, FGF21, and full-length (FL)-ANGPTL4, in *Ppara*[-/-] and *Ppara*[+/+] livers. U2AF65 and β-tubulin were used as loading controls for nuclear and cytoplasmic proteins, respectively. (**B–D**) Enrichment of the DNA fragment containing the PPAR response element (PPRE) (left panels) on the ACOX1 (**B**), Cyp4A14 (**C**), and FGF21 (**D**) promoters or their respective TSS (right panels) using anti-PPARα and anti-Pol2 antibodies or pre-immune IgG. Data are presented as mean ± SEM; n = 4–6, *p<0.05, **p<0.01, ***p<0.001 vs. untreated wild-type controls (two-tailed Mann-Whitney test).

involved in this pathway were up-regulated in parallel with a down-regulation of the rate-limiting genes involved in glycolysis (e.g., *Gck, Hk1*, and *Gpd2*) (*Figure 4—figure supplement 2D*). We also detected in the *Ppara*$^{-/-}$ liver a significantly elevated level of alanine aminotransferase (ALT), which is encoded by *Gpt* (*Figure 4—figure supplement 2C*). ALT is an enzyme that catalyzes the transfer of amino groups to form the hepatic metabolite oxaloacetate, an intermediate substrate in the tricar-boxylic acid (TCA)/Krebs cycle, which can be used for gluconeogenesis. Thus, we postulated that an increased anaplerotic oxidation of amino acids may be an alternative pathway engaged to maintain gluconeogenesis and to supply glucose as an energy source for normal growth in *Ppara*$^{-/-}$ neonates when the FAO capacity was impeded in the absence of PPARα. Indeed, most of the rate-limiting enzymes involved in amino acid oxidation (except for valine, leucine, and isoleucine) were concomi-tantly up-regulated in *Ppara*$^{-/-}$ liver at P2 (*Figure 4—figure supplement 2E*). Up-regulation of these genes would provide some of the intermediate substrates of the TCA cycle, such as α-ketoglutarate, fumarate, and oxaloacetate, which could be used for gluconeogenesis (*Figure 4—figure supple-ment 2E*). Importantly, selective inhibition of ALT by intraperitoneal injection of L-cycloserine led to decoupling of amino acid oxidation from gluconeogenesis and ultimately resulted in stunted postna-tal growth and hypoglycemia in *Ppara*$^{-/-}$ pups (*Figure 4—figure supplement 2F,G*). In short, we con-clude that PPARα functions both as a prenatally anticipatory and postnatally adaptive regulator of lipid catabolism, ultimately protecting the postnatal liver against a rapid insurgence of steatosis by promoting the use of milk lipids as an energy source.

## An epigenetic switch controls a subset of PPARα target genes

How PPARα target genes are controlled at different stages of metabolic ontogenesis remains unclear. For instance, the genes (e.g., *Acox1*) involved in the anticipatory functions of PPARα (i.e., FAO) are stimulated in the fetus, but the adaptive regulators of lipid catabolism, such as the liver-secreted FGF21, were markedly stimulated by PPARα only after birth (*Figures 2C,5A*). The binding of PPARα to the peroxisome proliferator response element (PPRE) in the promoter of *Acox1* and *Cyp4a14* (both genes are indicators for PPARα activation) (*Tugwood et al., 1992*; *Anderson et al., 2002*), as well as Pol2 recruitment to their respective TSS, occurs before birth and results in increased expression of these genes (*Figure 5C,D*). This stimulation correlates with higher *Ppara* TSS activity and mRNA levels during hepatic ontogenesis (*Figure 1B,C*). Despite having comparable PPARα occupancy in the *Fgf21, Acox1*, and *Cyp4a14* promoters before birth, the recruitment of Pol2 to the *Fgf21* TSS was markedly delayed until after birth at P2, and then gradually increased thereafter (*Figure 5D*).

Based on these observations, we postulated that other mechanisms may be involved in the differ-ential Pol2 recruitment and promoter transactivation of these PPARα target genes. At E19.5, we found increased active histone marks (i.e., AcH4 and H3K4me3) near the TSS of *Acox1* and *Cyp4a14*, which correlated with a reduced enrichment of repressive histone marks (i.e., H3K9me3 and H3K27me3) (*Figure 6A,B*). At P2, the enrichment of active histone marks at the TSS of *Acox1* and *Cyp4a14* decreased with a concomitant increase in repressive histone marks, but this was not observed for *Fgf21* (*Figure 6A–C*). HDAC3 has been reported to repress hepatic *Fgf21* expression (*Hondares et al., 2010*; *Estall et al., 2009*; *Archer et al., 2012*; *Feng et al., 2011*).In line with these reports, we observed a higher occupancy of HDAC3 at the *Fgf21* TSS at E19.5 compared to P2 (*Figure 6D*). We also found that the expression of *Angptl4* was also controlled by HDAC3 (*Figure 6E*). Interestingly, the lack of PPARα correlated with a ~10-fold increase in the HDAC3 occu-pancy at the *Fgf21* and *Angptl4* TSS (*Figure 6D,E*). In contrast, we detected negligible levels of HDAC3 recruitment to the *Cyp4a14* TSS (*Figure 6F*). Taken together, our results suggest that HDAC3-mediated histone modifications represent an additional mechanism by which certain PPARα target genes are temporally regulated.

## β-hydroxybutyrate induces *Fgf21* expression via HDAC3 inhibition

Previous studies indicate that milk lipids and hydroxymethylglutaryl-coenzyme A synthase 2 (HMGCS2) activity are required for *Fgf21* expression (*Hondares et al., 2010*; *Vilà-Brau et al., 2011*). We hypothesized that PPARα-dependent production of ketone bodies, particularly β-hydroxybuty-rate, which was ~seven-fold higher in neonates than fetuses (*Figure 4G*), may activate postnatal gene expression by alleviating the repressive role of class I HDACs (*Shimazu et al., 2013*).To test

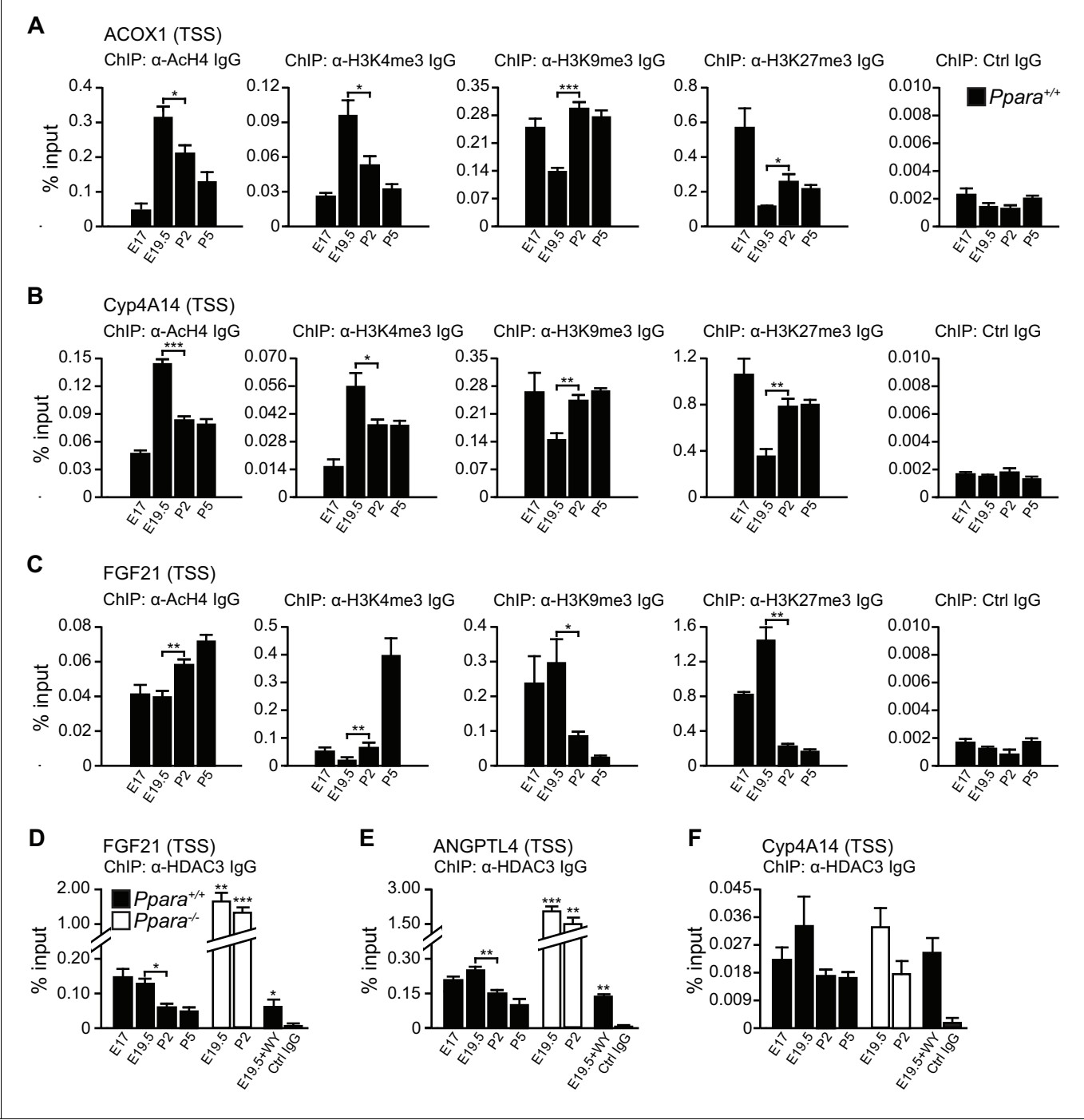

**Figure 6.** PPARα-mediated regulation of *Fgf21* expression is dictated by the occupancy of HDAC3 on its promoter. (A–C) Enrichment of the DNA fragment containing the ACOX1 (**A**), Cyp4A14 (**B**), or FGF21 (**C**) TSS in primary hepatocytes isolated from *Ppara*[+/+] pups using antibodies against acetyl-histone four (AcH4), trimethylated histone three at lysine four (H3K4me3), lysine nine (H3K9me3), lysine 27 (H3K27me3), or pre-immune IgG (Ctrl IgG) and evaluated by real-time qPCR. (D–F) Enrichment of the DNA fragment containing the FGF21 (**D**), ANGPTL4 (**E**), or Cyp4A14 (**F**) TSS in primary hepatocytes isolated from *Ppara*[-/-] and *Ppara*[+/+] pups with or without WY-14643 treatment in pregnant dams using anti-HDAC3 antibodies or control IgG. Data are presented as mean ± SEM; n = 4–6, *p<0.05, **p<0.01, ***p<0.001 vs. untreated wild-type controls unless otherwise indicated (two-tailed Mann-Whitney test).

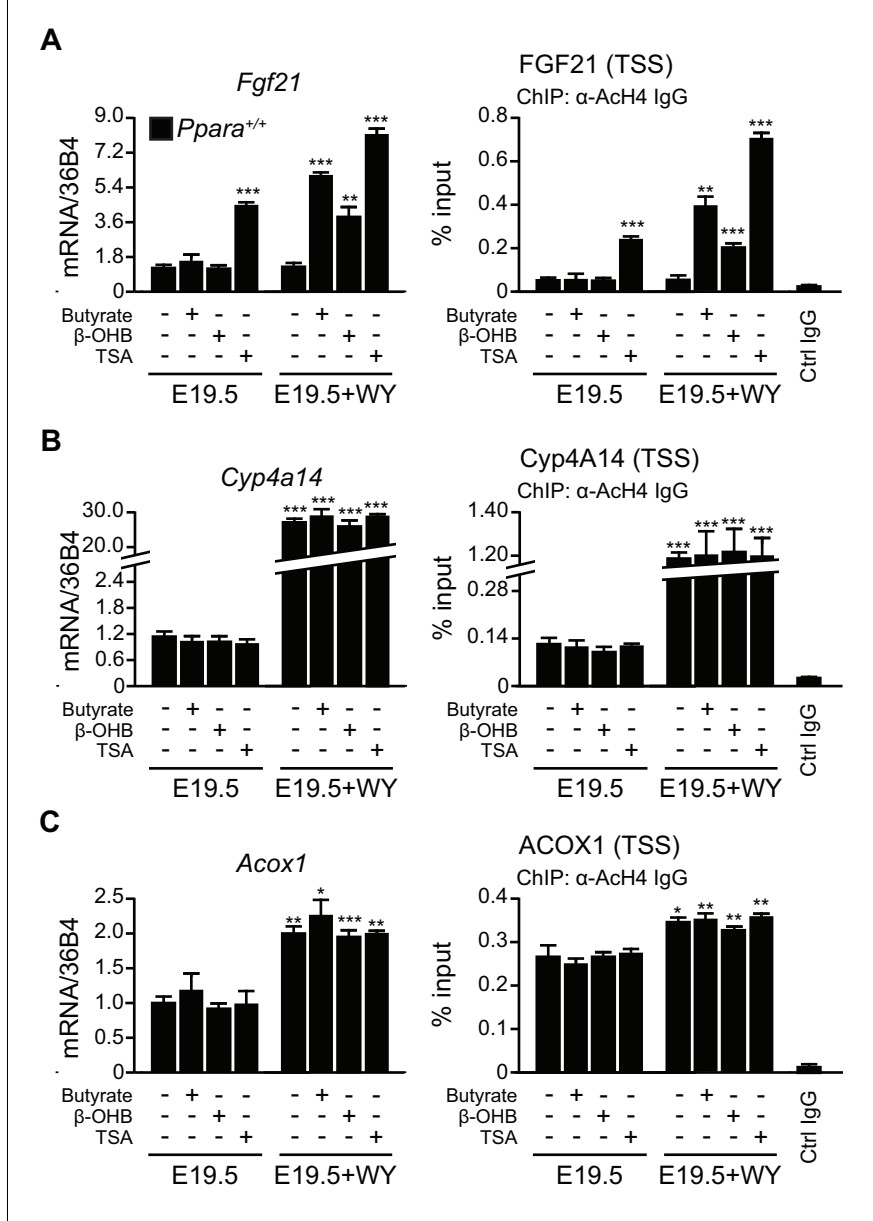

**Figure 7.** β-hydroxybutyrate acts as an endogenous inhibitor of HDAC3 to activate *Fgf21* expression upon milk suckling. (**A–C**) Ex vivo liver explants isolated from *Ppara*[+/+] fetuses at E19.5 were used to study the effects of butyrate, β-hydroxybutyrate, and trichostatin in the presence or absence of WY-14643 (WY) on PPARα target genes. *Left*: mRNA expression of *Fgf21* (**A**), *Cyp4a14* (**B**), and *Acox1* (**C**). *Right*: enrichment of the DNA fragment containing the FGF21 (**A**), Cyp4A14 (**B**), or ACOX1 (**C**) TSS after chromatin immunoprecipitation (ChIP) using antibody against acetyl-histone 4 (AcH4). Data are presented as mean ± SEM; n = 6, *p<0.05, **p<0.01, ***p<0.001 vs. E19.5 samples without WY-14643 treatment (two-tailed Mann-Whitney test).

this possibility, we treated *Ppara*[+/+]liver explants from E19.5 pups with β-hydroxybutyrate, sodium butyrate (a known HDAC3 inhibitor), and trichostatin A, a selective inhibitor of class I and class II HDACs. Activation of the *Fgf21* transcriptional activity was dependent on both the activation of PPARα and HDAC3 inhibition through butyrate or β-hydroxybutyrate treatment (*Figure 7A*). In particular, HDAC3 inhibition or PPARα activation alone did not cause a significant change in *Fgf21* induction. However, trichostatin A alone, regardless of PPARα activation, led to significantly higher expression of *Fgf21*, suggesting the involvement of a PPARα-independent pathway and other class I or class II HDACs in the regulation of *Fgf21* expression in addition to PPARα (*Figure 7A*). In contrast,

the activity of AcH4 at the TSS of *Cyp4a14* and *Acox1* and the expression of these genes were not influenced by any of the treatments except the PPARα ligand WY-14643, thereby excluding the involvement of HDAC3 or other class I and class II HDACs in the regulation of their transcriptional activity (*Figure 7B,C*). These results suggest that milk lipids can affect the epigenetic status of the postnatal liver in which hepatic production of β-hydroxybutyrate acts as an inhibitor of HDAC3 that regulates *Fgf21*.

## Discussion

In mammals, the transition to extra-uterine life represents a sudden shift in the source of energy (i.e., from a carbohydrate-laden fetal diet to a high-fat, low-carbohydrate diet). The neonatal liver must coordinate hepatic FAO, gluconeogenesis, and ketogenesis in order to maintain bioenergetic homeostasis and to meet the metabolic demands associated with extra-uterine life which depends exclusively on milk (*Cotter et al., 2013*). In contrast to the carbohydrate-replete nutrient state experienced in utero, neonatal energy is dominated by PPARα-dependent transcriptional regulation of lipid catabolism. We found that this anticipatory regulation of PPARα activity and lipid metabolism is directly controlled by GR before birth.

Glucocorticoid hormones are crucial in the functional maturation of many key tissues, notably in preparation for birth (*Cottrell and Seckl, 2009*). The initiation of parturition is marked by short bursts of stress hormones (*Barlow et al., 1974*). We demonstrate that this short burst of glucocorticoids at the initiation of birth directly stimulate the GR-dependent transcription of PPARα and its lipid catabolic target genes, which in turn prepare neonates for the sudden shift to fat-rich milk diet as the primary source of energy (*Figure 8A,B*). Indeed, we previously demonstrated the GR-mediated hormonal induction of PPARα in adult hepatocytes (*Lemberger et al., 1994*), while others showed that ligand-activated PPARα interferes with the recruitment of GR and Pol2 to the promoter of classical GRE-driven genes, inhibiting their transcription (*Bougarne et al., 2009*). This is also consistent with our observation of a reduced expression of *Nr3c1* in *Ppara*$^{-/-}$ livers, implicating a reciprocal regulatory relationship between GR and PPARα.

We indicate that lipid catabolic target genes of PPARα, particularly those involved in mitochondrial, peroxisomal, and microsomal oxidation, are switched on just before birth in anticipation of a postpartum lipid-rich meal. In the neonatal liver, however, we show that mitochondrial β-oxidation prevails as the major contributor to the neonatal oxidative capacity and protects against hepatic steatosis when faced with a sudden surge in dietary fat. Our findings are consistent with the notion that mitochondrial FAO represents the dominant metabolic pathway whereas peroxisomal FAO assumes a relatively minor role (*Hashimoto et al., 1999*). Accordingly, long-chain FAs constitute more than 60% of the total FAs in animals and milk (*Smith et al., 1968*), and their abundance makes them the only significant source of metabolic fuel for the mitochondrial, but not peroxisomal, β-oxidation system (*Reddy and Mannaerts, 1994*). Moreover, decreased mitochondrial FAO has been considered one of the major contributing factors leading to liver steatosis (*Ockner et al., 1993*). Thus, we conclude that the concomitant presence of lipid content of milk, a lower number of mitochondria and overt mitochondrial dysfunctions jointly contributes to the development of liver steatosis in *Ppara*$^{-/-}$neonates. Notably, mitochondrial dysfunctions were evident in *Ppara*$^{-/-}$ hepatocytes due to reduced $\Delta_{\Psi M}$ and increased intracellular ROS levels. We postulate that an increased intracellular ROS level likely precedes the loss of $\Delta_{\Psi M}$ and mitochondria dysfunction. As this oxidative stress condition continues to worsen after birth, perhaps induced by nutritional lipid as previously indicated (*Tirosh et al., 2009*), reduced $\Delta_{\Psi M}$ and ATP production ultimately ensue. This is in agreement with a previous report of a similar mitochondrial disorder associated with the dysfunctional respiratory chain components (*Lebiedzinska et al., 2010*). Particularly, the increased intracellular ROS levels observed in *Ppara*$^{-/-}$ hepatocytes may be due to the reduced clearance of ROS as evidenced by the down-regulation of genes coding for mitochondrial superoxide dismutase 2 (SOD2) and copper chaperone for SOD1 (a metalloprotein responsible for delivering copper as a co-factor to SOD1) before and/or after birth (*Figure 2—source data 1*). This observation is consistent with the report that PPARα activation by agonist clofibrate stimulates SOD1 and catalase activities as part of the defense mechanisms against oxidative stress in the heart (*Ibarra-Lara et al., 2016*).

Many of the PPARα-regulated genes involved in FAO have previously been shown to contain a PPRE in their promoter region and hence, recognized as direct PPARα target genes

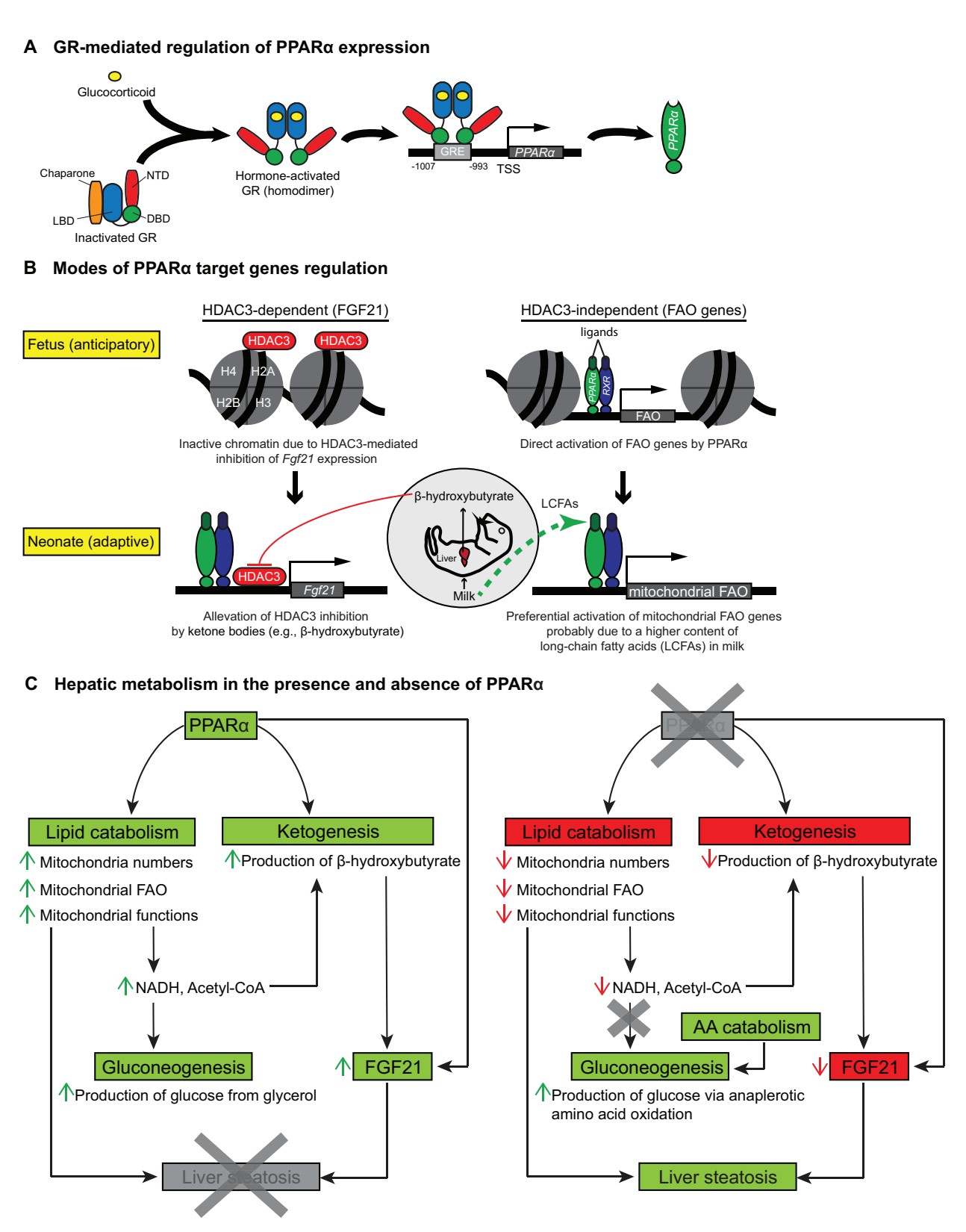

**Figure 8.** Schematic illustrations of GR-mediated regulation of PPARα expression, HDAC3-dependent and -independent regulation of PPARα target genes before and after birth, and the affected liver metabolic processes with or without PPARα. (**A**) In fetal liver, hormone-activated GR binds to the GR

*Figure 8 continued*

response element (GRE) spanning -1007 to -993 within the PPARα promoter region to directly activate the transcription of *Ppara*. NTD: N-terminal domain, LBD: ligand-binding domain, DBD: DNA-binding domain. (**B**) At E19.5, direct binding of HDAC3 near the TSS of *Fgf21* leads to the repression of *Fgf21* transcriptional activity. Upon milk suckling, PPARα-dependent production of β-hydroxybutyrate from the neonatal liver alleviates the HDAC3-mediated repression of *Fgf21* by directly inhibiting the activity of HDAC3, permitting PPARα-dependent *Fgf21* expression. Activated GR leads to the stimulation of PPARα and its target genes in fetal liver, such as those involved in mitochondrial and peroxisomal fatty acid oxidation (FAO). After birth, the expression of PPARα target genes involved in mitochondrial FAO predominates the expression of genes involved in peroxisomal FAO, probably due to the higher long-chain fatty acid (LCFA) content than very-long-chain fatty acid (VLCFA) content in the milk. (**C**) Schematic illustration depicting the affected processes of hepatic metabolism in the presence and absence of PPARα. Green boxes/arrows indicate increments; reds indicate decrements.

(Rakhshandehroo et al., 2010). This implicates that the reduced oxidative capacity observed in *Ppara*[-/-] mice was due to the lack of direct PPARα-mediated stimulatory effects rather than a cellular-autonomous metabolic adaptation in response to lower mitochondria numbers. As discovered by Semenkovich and *Rakhshandehroo et al., 2010* co-workers (*Chakravarthy et al., 2005*, *2009*), newly synthesized FA by fatty acid synthase may also regulate glucose, lipid, and cholesterol metabolism by serving as endogenous activators of a distinct physiological pool of PPARα ligands in adult liver. However, we doubt that de novo synthesis of FA plays a predominant role in the regulation of PPARα activity during early postnatal period since insulin/glucagon ratio is low, which promotes FAO rather than de novo synthesis of FA (*Girard et al., 1977*). Our recent work indicates that PPARα has little impact on the expression of lipogenic genes in normal conditions (*Montagner et al., 2016*). Moreover, an effective hepatic FAO is essential for the provision of acetyl-CoA and NADH to support hepatic de novo ketogenesis and gluconeogenesis (*Girard, 1986*). This notion is consistent with the hypoketonemia and hypoglycemia in *Ppara*[-/-] neonates reported in this work and by Crawford and co-workers (*Cotter et al., 2014*). Notably, the hypoglycemic phenotype previously reported in *Ppara*[-/-] neonates by Cotter et al. happened at P1 due to a decreased hepatic gluconeogenesis from glycerol. However, we did not detect any sign of hypoglycemia in these mice starting from P2. This apparent discrepancy is likely attributable to the rapid (~1 day) and dynamic adaptive functions of *Ppara*[-/-] neonatal liver in providing glucose as the primary source of energy required for postnatal growth and in maintaining glucose homeostasis. Our finding also supports the work of others indicating a preserved gluconeogenesis from glycerol in adult *Ppara*[-/-] mice (*Xu et al., 2002*). We elucidated that this cell-autonomous metabolic adaptation was enhanced in the absence of PPARα and mediated through the up-regulation of hepatic ALT and anaplerotic amino acid oxidation. Thus far, the mechanism behind the increased production and secretion of ALT in *Ppara*[-/-] neonatal liver still remains unclear. It is notable that *ALT2* expression is up-regulated under metabolic stress and that plasma and tissue ALT levels are elevated in response to endoplasmic reticulum (ER) stress (*Salgado et al., 2014*; *Josekutty et al., 2013*). Interestingly, defective PPARα signaling has been reported to cause hepatic mitochondrial and ER stress in the pathogenesis of hepatic steatosis through reduced mRNA expression of the sarco/endoplasmic reticulum calcium ATPase (SERCA) (*Su et al., 2014*). In addition, our transcriptome analysis revealed a consistent reduction by ~1.2-fold in the expression of *Atp2a2*, which encodes SERCA, in *Ppara*[-/-] liver at E19.5 and P2 (*Figure 2—source data 1*). Moreover, we also delineated that most of the rate-limiting enzymes involved in amino acid oxidation were concomitantly up-regulated in *Ppara*[-/-] neonatal liver to provide essential intermediate substrates in TCA/Krebs cycle required for gluconeogenesis. Hence, we speculate that de novo synthesis of glucose becomes the primary energy source in sustaining normal postnatal growth in *Ppara*[-/-] neonates as evidenced by the observed growth retardation upon decoupling of amino acid oxidation from gluconeogenesis. This is in conjunction with the failure of *Ppara*[-/-] neonatal liver in harnessing energy from ketone bodies and lipids due to defective mitochondrial functions. In parallel, the compensatory up-regulation of pyruvate carboxylase (Pcx) and other gluconeogenic genes in *Ppara*[-/-] neonates at P2 probably corresponds with a lower hepatic acetyl-CoA level, as supported by a recent report that hepatic acetyl-CoA acts as an allosteric regulator of Pcx (*Perry et al., 2015*). Importantly, these findings in *Ppara*[-/-] mice illustrate the tremendous flexibility of the neonatal liver in coordinating ketogenesis, FAO, and gluconeogenesis so as to meet the energy demand required for normal growth especially when faced with an energy crisis caused by defective oxidative metabolism and hypoketonemia.

Mitochondrial 3-hydroxy-3-methylglutaryl-CoA synthase 2 (HMGCS2) is one of the rate-liming enzymes involved in ketogenesis. *Hmgcs2* contains a PPRE in its promoter region and is under the direct transcriptional control of PPARα (*Rodríguez et al., 1994*). HMGCS2 has been shown to induce mitochondrial FAO and FGF21 expression, possibly via a SIRT1-dependent mechanism (*Vilà-Brau et al., 2011*). Furthermore, *Hmgcs2* mRNA expression peaks 16 hr after birth and coincides with a decline in *Acox1* mRNA expression thereafter (*Yubero et al., 2004*; *Serra et al., 1993*, *1996*). These reports substantiate our findings of preferential activation of mitochondrial FAO at birth. Our results also support the in vivo finding that β-hydroxybutyrate is a potent physiological inhibitor of HDAC3 (*Shimazu et al., 2013*), suggesting that PPARα-mediated activation of HMGCS2 and β-hydroxybutyrate production precedes the PPARα-regulated production of FGF21. Similar to other PPARα target genes, we detected PPARα occupancy in the *Fgf21* promoter before birth, but its transcription is ultimately determined by the alleviation of HDAC3-mediated inhibition (*Figure 8B*). Thus, fatty acid catabolism provides β-hydroxybutyrate, which acts as a secondary signal to initiate the de-repression of the *Fgf21* promoter by HDAC3, allowing it to become PPARα responsive. It was previously indicated that HDAC-mediated histone deacetylation may inhibit transcription at the initiation and/or elongation step (*Wang et al., 2009*). If the genes are loaded by Pol2 at the TSS, similar to what we observed at the *Fgf21* TSS upon PPARα agonistic activation, HDAC-mediated inhibition of transcription most likely happens at the elongation step although abortive initiation of the TSS bound Pol2 cannot be excluded. Our results suggest that PPARα activation induces the recruitment and assembly of a transcription initiation complex at the *Fgf21* TSS. However, the elongation of *Fgf21* mRNA transcript only ensues after alleviation of HDAC3 inhibition by β-hydroxybutyrate. Therefore, it is consistent with our findings that PPARα agonist has no effect on *Fgf21* mRNA or transcriptional activation in the absence of β-hydroxybutyrate.

Both human and mouse neonates exhibit mild ketogenic conditions during early postpartum period (0–1 day, 0.2–0.5 mmol/L), which further exacerbates after 5–10 days (0.5–1.1 mmol/L) (*Hamosh, 2004*). We believe that it could be due to the use of lipids as preferred fuel for growth while the carbohydrate parts of milk (e.g., lactose and oligosaccharide—two main carbohydrates in milk) mainly contribute to brain growth (i.e., myelin synthesis) and gut microbiota nourishment (*Edmond, 1992*; *Chichlowski et al., 2011*). Further, the observed ketogenic condition may also be explained by the low insulin/glucagon ratio in early postpartum life (*Girard et al., 1977*; *Ferré et al., 1979*; *Girard et al., 1992*). We found that ketone bodies act as a hit that is important to prime PPARα-dependent expressions of *Fgf21* and *Angptl4* by modulating HDAC3 activity on their promoter. Both FGF21 and ANGPTL4 function as secreted systemic effectors of PPARα (*Badman et al., 2007*; *Dijk and Kersten, 2014*). However, unlike *Fgf21*, the expression of *Angptl4* has already begun in the fetal liver. This finding is congruent with the report that ANGPTL4 functions as a glucocorticoid-dependent gatekeeper of FA flux by inhibiting the activity of adipose tissue lipoprotein lipase during fasting (*Koliwad et al., 2012*; *Dijk and Kersten, 2014*). In fact, significant overlap exists between the bioenergetic challenges experienced during fasting and at birth. Liver-derived FGF21 acts as an endocrine regulator of ketogenesis, which links the phenotypes observed during fasting and in newborns (*Badman et al., 2007*; *Hondares et al., 2010*). This concept is evident in *Ppara*[-/-] mice, which have reduced plasma FGF21 levels. During fasting, the switch to a lipid-dominated nutrient supply provokes hypoketonemia, hypoglycemia, and hepatic steatosis in adult *Ppara*[-/-] mice (*Kersten et al., 1999*; *Montagner et al., 2016*). In concordance with a recent report (*Cotter et al., 2014*), the shift to a high-fat, low-carbohydrate ketogenic diet in *Ppara*[-/-] neonates results in steatosis similar to that observed in fasted adult *Ppara*[-/-] mice. Notably, FGF21 has previously been shown to stimulate hepatic FAO and to prevent hepatic steatosis following ingestion of lipid-laden milk (*Xu et al., 2009*). Hence, we believe that the physiological relevance of FGF21 induction in neonates after milk suckling relates to its role in metabolic regulation by stimulating lipid catabolism rather than a response to starvation or a low calorie condition which no longer exists upon milk ingestion. It is also important to note that the early human breast milk is not initially ketogenic due to the relatively high lactose content, which gradually decreases while the fat content increases as the milk matures over time (*Jenness, 1979*). In contrast, mouse milk is likely a ketogenic diet as evidenced by a previous report that early mouse milk comprises ~17% of fat and ~1.7% of lactose at P3 (*Görs et al., 2004*). Therefore, direct translation of our findings to human situations needs to be cautioned for such differences in milk composition.

**Table 1.** Standard chow diet formulation.

**Manufacturer**: SPECIALITY FEEDS
**Product ID**: AIN93M

|  Standard AIN93M rodent diet  |
|---|

A semi-pure diet formulation for laboratory rats and mice based on AIN-93M. This formulation satisfies the maintenance nutritional requirements of rats and mice. Some modifications have been made to the original formulation to suit locally available raw materials.

- We have evidence that vitamin losses and other changes to the diet can occur when irradiated at 25KGy. The diet SF08-020 has been formulated for irradiation. Please contact us for more information if the diet is to be irradiated.

| Calculated nutritional parameters | | Ingredients | |
|---|---|---|---|
| Protein | 13.6% | Casein (acid) | 140 g/kg |
| Total fat | 4.0% | Sucrose | 100 g/kg |
| Total digestible carbohydrate as defined by FSANZ Standard 1.2.8 | 64.8% | Canola oil | 40 g/kg |
| | | Cellulose | 50 g/kg |
| Crude fiber | 4.7% | Wheat starch | 472 g/kg |
| AD fiber | 4.7% | Dextrinised starch | 155 g/kg |
| Digestible energy | 15.1 MJ/kg | DL methionine | 1.8 g/kg |
| % Total calculated digestible energy from lipids | 9.0% | Calcium carbonate | 13.1 g/kg |
| % Total calculated digestible energy from protein | 15.0% | Sodium chloride | 2.6 g/kg |
| | | AIN93 trace minerals | 1.4 g/kg |
| Diet form and features | | Potassium citrate | 1.0 g/kg |
| | | Potassium dihydrogen phosphate | 8.8 g/kg |
| - Semi pure diet. 12 mm diameter pellets. | | | |
| - Pack size 5 kg. Vacuum packed in oxygen impermeable plastic bags, under nitrogen. Bags are packed into cardboard cartons for protection during transit. Smaller pack quantity on request. | | Potassium sulphate | 1.6 g/kg |
| | | Choline chloride (75%) | 2.5 g/kg |
| - Diet suitable for irradiation but not suitable for autoclave. | | AIN93 vitamins | 10 g/kg |
| - Lead time 2 weeks for non-irradiation or 4 weeks for irradiation. | | | |
| Calculated amino acids | | Calculated total vitamins | |
| Valine | 0.90% | Vitamin A (retinol) | 4000 IU/kg |
| Leucine | 1.30% | Vitamin D (cholecalciferol) | 1000 IU/kg |
| Isoleucine | 0.60% | Vitamin E (α-tocopherol acetate) | 75 mg/kg |
| Threonine | 0.60% | Vitamin K (menadione) | 1 mg/kg |
| Methionine | 0.60% | Vitamin C (ascorbic acid) | None added |
| Cystine | 0.05% | Vitamin B1 (thiamine) | 6.1 mg/kg |
| Lysine | 1.00% | Vitamin B2 (riboflavin) | 6.3 mg/kg |
| Phenylanine | 0.70% | Niacin (nicotinic acid) | 30 mg/kg |
| Tyrosine | 0.70% | Vitamin B6 (pryridoxine) | 7 mg/kg |
| Tryptophan | 0.20% | Pantothenic acid | 16.5 mg/kg |
| Histidine | 0.42% | Biotin | 200 µg/kg |
| Calculated total minerals | | Folic acid | 2 mg/kg |
| Calcium | 0.47% | Inositol | None added |
| Phosphorus | 0.35% | Vitamin B12 (cyancobalamin) | 103 µg/kg |
| Magnesium | 0.09% | Choline | 1670 mg/kg |
| Sodium | 0.15% | Calculated fatty acid composition | |
| Chloride | 0.16% | Myristic acid 14:0 | No data |
| Potassium | 0.40% | Palmitic acid 16:0 | 0.20% |
| Sulphur | 0.17% | Stearic acid 18:0 | 0.10% |
| Iron | 75 mg/kg | Palmitoleic acid 16:1 | No data |

*Table 1 continued on next page*

*Table 1 continued*

**Manufacturer**: SPECIALITY FEEDS
**Product ID**: AIN93M

| Standard AIN93M rodent diet | | | |
|---|---|---|---|
| Copper | 6.9 mg/kg | Oleic acid 18:1 | 2.40% |
| Iodine | 0.2 mg/kg | Gadoleic acid 20:1 | trace |
| Manganese | 19.5 mg/kg | Linoleic acid 18:2 n6 | 0.80% |
| Cobalt | No data | α-Linolenic acid 18:3 n3 | 0.40% |
| Zinc | 47 mg/kg | Arachidonic acid 20:4 n6 | No data |
| Molybdenum | 0.15 mg/kg | EPA 20:5 n3 | No data |
| Selenium | 0.3 mg/kg | DHA 22:6 n3 | No data |
| Cadmium | No data | Total n3 | 0.45% |
| Chromium | 1.0 mg/kg | Toal n6 | 0.76% |
| Fluoride | 1.0 mg/kg | Total mono-unsaturated fats | 2.46% |
| Lithium | 0.1 mg/kg | Total polyunsaturated fats | 1.21% |
| Boron | 3.1 mg/kg | Total saturated fats | 0.28% |
| Nickel | 0.5 mg/kg | | |
| Vanadium | 0.1 mg/kg | | |

Calculated data uses information from typical raw material composition. It could be expected that individual batches of diet will vary from this figure. Diet post treatment by irradiation or autoclave could change these parameters. We are happy to provide full calculated nutritional information for all of our products, however we would like to emphasise that these diets have been specifically designed for manufacture by Specialty Feeds.

In summary, we provide evidence that prenatal expression of PPARα is under the direct control of GR. The GR-dependent PPARα activity is pivotal for the induction of hepatic FA catabolic genes, pointing to a novel anticipatory role of PPARα in the fetal liver in preparation for the efficient use of milk fat as an energy source. In addition, hepatic FA catabolism provides essential co-factors for the synthesis of ketone bodies such as β-hydroxybutyrate. Interestingly, ketone bodies act as a secondary signal that further activates PPARα-dependent regulation of the hepatokine FGF21. Therefore GR-PPARα axis may represent a critical signaling pathway in late gestation for the ability of mammalian newborns to use nutrients and maintain whole-body homeostasis. We conclude that in the absence of PPARα, a vicious cycle pertaining to the (i) lower mitochondria numbers, (ii) mitochondrial dysfunctions, (iii) impaired mitochondrial FA β-oxidation, (iii) hypoketonemia either due to a lack of direct PPARα stimulation of *Hmgcs2* or a lack of essential co-factors provided by functional FAO, (iv) lack of epigenetic activation of FGF21 by ketone bodies (e.g., β-hydroxybutyrate), and (v) a surge in nutritional lipids upon milk suckling, exists to concomitantly contribute to the development of hepatic steatosis in neonates (*Figure 8C*).

## Materials and methods

### Animals

PPARα-null (*Ppara*[-/-]) and GR-null (*Nr3c1*[-/-]) mice were acquired from the Jackson Laboratory (Bar, Harbor, ME) and Nuclear Receptor Zoo (MGI: 95824, Strasbourg, France), respectively. These mice were given standard rodent chow diet (AIN93M, Specialty Feeds, Australia) (*Table 1*) and water ad libitum, maintained in a C57BL/6 background, and bred in our specific pathogen-free facilities by inter-crossing *Ppara*[+/-] mice to obtain experimental wild-type and knockout pups in the same litter. This breeding strategy allowed the experimental mice to be exposed to the same gestational environment and ensured that pups received milk of the same nutritional content from the heterozygous dams. Pregnancy was timed based on vaginal plug formation. Pregnant dams approaching full-term were closely monitored and cesarean section performed to obtain fetuses at E19.5 or earlier (E13, E15, E17). Pups at P2 were timed based on the delivery day. Litters of 6–8 pups were used to

**Table 2.** Primer sequences for mouse genes used in quantitative real-time PCR.

| Gene | Forward primer (5'-3') | Reverse primer (5'-3') |
|---|---|---|
| Acaa2 | ATGTGCGCTTCGGAACCAAA | CAAGGCGTATCTGTCACAGTC |
| | Acetyl-coenzyme A acyltransferase 2/short chain-specific 3-ketoacyl-CoA thiolase (mitochondrial) | |
| Acadl | TGCCCTATATTGCGAATTACGG | CTATGGCACCGATACACTTGC |
| | Acyl-coenzyme A dehydrogenase, long chain | |
| Acadvl | TGACCTTGGTGTTAGCGTTAC | CTGGGCCTTTGTGCCATAGAG |
| | Acyl-coenzyme A dehydrogenase, very long chain | |
| Acox1 | TCGAAGCCAGCGTTACGAG | ATCTCCGTCTGGGCGTAGG |
| | Acyl-CoA oxidase 1, palmitoyl | |
| Acsl1 | ACCAGCCCTATGAGTGGATTT | CAAGGCTTGAACCCCTTCTG |
| | Acyl-CoA synthetase, long chain family member 1 | |
| Angptl4 | TCCAACGCCACCCACTTAC | TGAAGTCATCTCACAGTTGACCA |
| | Angiopoietin-like 4 | |
| Cpt1a | CTATGCGCTACTCGCTGAAGG | GGCTTTCGACCCGAGAAGA |
| | Carnitine palmitoyltransferase 1a (liver) | |
| Cpt2 | CAAAAGACTCATCCGCTTTGTTC | CATCACGACTGGGTTTGGGTA |
| | Carnitine palmitoyltransferase 2 | |
| Cyp4a14 | TCATGGCGGACTCTGTCAATA | GCAGGCGAAAGAAAGTCAGG |
| | Cytochrome P450, family 4, subfamily a, polypeptide 14 | |
| Ehhadh | ACAGCGATACCAGAAGCCAG | TGGCAATCCGATAGTGACAGC |
| | Enoyl-coenzyme A, hydratase/3-hydroxyacyl coenzyme A dehydrogenase | |
| Fabp1 | AAGGCAGTCGTCAAGCTGG | CATTGAGTTCAGTCACGGACTT |
| | Fatty acid bind protein 1 (liver) | |
| Fgf21 | CTGCTGGGGGTCTACCAAG | CTGCGCCTACCACTGTTCC |
| | Fibroblast growth factor 21 | |
| Gck | AGACGAAACACCAGATGTATTCC | GAAGCCCTTGGTCCAGTTGAG |
| | Glucokinase | |
| Nr3c1 | CCGGGTCCCCAGGTAAAGA | TGTCCGGTAAAATAAGAGGCTTG |
| | Glucocorticoid receptor | |
| Hadha | AGCAACACGAATATCACAGGAAG | AGGCACACCCACCATTTTGG |
| | Hydroxyacyl-coenzyme A dehydrogenase, alpha subunit | |
| Hadhb | TGAATATGCACTGCGTTCTCAT | CCTTTCCTGGTACTTTGAAGGG |
| | Hydroxyacyl-coenzyme A dehydrogenase, beta subunit | |
| Hk1 | CGGAATGGGGAGCCTTTGG | GCCTTCCTTATCCGTTTCAATGG |
| | Hexokinase 1 | |
| Fkbp51 | TTTGAAGATTCAGGCGTTATCCG | GGTGGACTTTTACCGTTGCTC |
| | FK506 binding protein 51 | |
| Pex19 | GACAGCGAGGCTACTCAGAG | GCCCGACAGATTGAGAGCA |
| | Peroxisomal biogenesis factor 19 | |
| Ppara | TCGGCGAACTATTCGGCTG | GCACTTGTGAAAACGGCAGT |
| | Peroxisome proliferator activated receptor alpha | |
| Slc25a20 | GCGCCCATCATTGGAGTCA | CACACCAGATAACATCCCAGC |
| | Solute carrier family 25 (mitochondrial carnitine/acylcarnitine translocase), member 20 | |
| 36B4/RplP0 | CGAGGACCGCCTGGTTCTC | GTCACTGGGGAGAGAGAGG |
| | Ribosomal protein P0 | |

minimize differences in milk availability verified by stomach inspection. All experimental protocols involving animals were reviewed and approved by the Veterinary Office of the Canton Vaud (SCA–EXPANIM, Service de la Consommation et des Affaires Vétérinaires, Epalinges, Switzerland) in accordance with the Federal Swiss Veterinary Office Guidelines and by the Institutional Animal Care and Use Committee (#2013/SHS/866) in Singapore. The animal handling procedures were compliant with the *NIH Guide for the Care and Use of Laboratory Animals.*

## Tissue RNA isolation and quantitative real-time PCR

Total RNA was extracted from liver samples using TRIzol reagent (Life Technologies, Carlsbad, CA) and purified using RNeasy Mini Kit (Qiagen, Hilden, Germany) according to the manufacturer's instructions. The purified RNA was spectrophotometrically quantified and its quality assessed by measuring the absorbance ratios at 260 nm/280 nm and 260 nm/230 nm using Nanodrop Spectrophotometer (Thermo Fisher Scientific, Wilmington, DE). One microgram of mRNA was reverse-transcribed to cDNA using Superscript II Reverse Transcriptase (Life Technologies, Carlsbad, CA). The cDNA template was amplified by real-time PCR using iTaq SYBR Green Supermix (Bio-Rad, Hercules, CA). Relative mRNA levels were calculated using the comparative $2^{-\Delta\Delta CT}$ method after normalization to *36B4/RplP0* expression, which was used as an invariant control. The primer sequences used for real-time PCR were obtained from the Harvard PrimerBank (http://pga.mgh.harvard.edu/primerbank) and are provided in *Table 2*.

## ChIP assays

Eight to ten livers were pooled, homogenized, and cross-linked with 1% formalin at room temperature for 5 min. The reaction was stopped with the addition of glycine and nuclei isolated by sucrose-density ultracentrifugation. The nuclei were sonicated to produce DNA fragments of ~500–1000 base pairs. Total chromatin was incubated with 10 µg of antibody overnight at 4°C and precipitated with 10 µl of blocked protein A-agarose beads (Life Technologies, Carlsbad, CA) at 4°C for 2 hr. After de-crosslinking, the DNA was purified and the enrichment quantified by quantitative real-time PCR, expressing the results as the percentage of input. Antibodies against RNA polymerase II (Pol2, #sc-67318X), GR (#sc-1004X), and pre-immune rabbit antibodies (#sc-2027X) were purchased from Santa Cruz Biotechnology (Santa Cruz, CA). Antibodies against PPARα (#AB2779), histone deacetylase 3 (HDAC3, #AB7030), acetyl-histone 4 (AcH4), and trimethylated lysine 4 (H3K4me3, #AB8580), lysine 9 (H3K9me3, #AB8898), and lysine 27 (H3K27me3, #AB6002) of histone 3 were from Abcam (Cambridge, MA). The primer sequences used for chromatin immunoprecipitation were provided in *Table 3*.

**Table 3.** Primer sequences for mouse genes used in chromatin immunoprecipitation.

| Gene | Forward primer (5'-3') | Reverse primer (5'-3') |
|---|---|---|
| ACOX1_TSS | TCCCGGAAAGATCACGTGAACC | TCCCCGAGCGGCTCCTCGCCA |
| ACOX1_PPRE | TAGCCAACGACAATGAACC | CGGAAACCAGAAGGGAATG |
| ANGPTL4_TSS | CCAGCAAGTTCATCTCGTCC | TCCCTCCCACTCCCACACC |
| CYP4A14_TSS | ATTCCCCCTCCCACAAGTAG | CCCATGGTTAGTAGTTTCTGGA |
| CYP4A14_PPRE | AAGGAAAAGGCCACCGTCTA | TCCATCTCACTGAACTTTACCC |
| FGF21_TSS | ATATCACGCGTCAGGAGTGG | TCCCCAGCTGAGAAGACACT |
| FGF21_PPRE | AGGGCCCGAATGCTAAGC | AGCCAAGCAGGTGGAAGTCT |
| PPARα_TSS | GTTGTCATCACAGCTTAGCG | CAGATAAGGGACTTTCCAGGTC |
| PPARα_GRE (−1007 to −993) | GGGACTCGGGGAACAAGCTGTGCGATCTAG | GGAAGGGTGCGCCTTGGCGCGCACTCC |
| PPARα_GRE (−2080 to −2066) | CTTTCCTCTCAATACAGTCTGTCAAACAAAA | GTTTTGTTTGTTTTGACTCTCTGTCCAG |
| PPARα_GRE (−2953 to −2939) | AAGGGTGAACACACTTTGTTTCCTGGATG | TCCACCAGGGCAGGGGAAGTAGGTATT |

## Dexamethasone treatment

E15 fetal liver explants were used for treatment with dexamethasone due to their relatively low PPARα activation status. Fresh liver sections (n = 6 per treatment group) were cultured in the presence of dexamethasone (0.1 μM, 1 μM, or 10 μM) for 24 hr. After treatment, total RNA was isolated and purified, and PPARα mRNA levels were measured by quantitative real-time PCR.

## Primary hepatocyte isolation and culture

Fetuses or pups were sacrificed by decapitation. The liver was removed, minced into fine consistency with a pair of scissors, and digested at 37°C with agitation at 115 rpm for 1 hr in 10 ml of sterile-filtered 4-(2-hydroxyethyl)-1-piperazineethanesulphonic acid (HEPES):collagenase solution containing 0.1 M HEPES, 0.12 M NaCl, 50 mM KCl, 5 mM D-glucose, 1.5% bovine serum albumin (BSA), 1 mM CaCl$_2$, and 10,000 activity units of type-I collagenase (Life Technologies, Carlsbad, CA). The cell suspension was filtered through a 100-μm nylon mesh (BD Falcon, Franklin Lakes, NJ) and collected in a 50-ml centrifuge tube. The suspension was centrifuged at 200 × g for 10 min to pellet the cells and the supernatant removed. Red blood cell (RBC) lysis solution was added to re-suspend the cells, followed by incubation at room temperature for 5 min. After RBC lysis, ice-cold PBS was added and the cell suspension filtered through a 40-μm nylon mesh (BD Falcon, Franklin Lakes, NJ) to remove cell clumps. Finally, the cell suspension was centrifuged at 200 ×g for 5 min to pellet the cells. For primary hepatocyte culture, the isolated mouse hepatocytes (2 × 10$^5$ cells/per 9.5 cm$^2$ well) were then cultured in DMEM containing 10% fetal calf serum (FCS) and 2 mM penicillin/streptomycin in 6-well plates coated with rat tail collagen (Corning, Tewksbury, MA).

## siRNA-mediated knockdown of *Nr3c1*

Primary hepatocytes were isolated and cultured as described above. Upon reaching a confluency of ~60%, the adherent cells were treated with 50 ng ON-TARGETplus NR3C1/GR-targeting SMART-pool siRNAs (L-045970-01-0005) or non-targeting pool (D-001810-10-05) (GE Dharmacon, Lafayette, CO) for 24 hr using transfection reagent DharmaFECT 1 as per the manufacturer's protocol. Cells were harvested for α-GR ChIP assays after 48 hr of incubation with complete medium. Successful knockdown was determined by reduced *Nr3c1* mRNA expression of at least 80% when compared with non-targeting siRNA treatment group.

## Gene expression microarray and analysis

RNA samples were randomly selected from three different litters for each genotype (n = 6). Total RNA was extracted from mouse liver using TRIzol reagent (Life Technologies, Carlsbad, CA) and further purified using RNeasy Mini Kit (Qiagen, Hilden, Germany) according to the manufacturer's instructions. RNA quality measurements were performed using an Agilent 2100 bioanalyzer (Agilent Technologies, Waldbronn, Germany). Only samples with intact bands corresponding to the 18S and 28S rRNA subunits without contamination with chromosomal DNA and had an RNA integrity number > 8.0 were selected for array hybridization. Hybridization was performed with Affymetrix Mouse Genome MoGene1.0 ST arrays according to the manufacturer's protocol and analyzed as described previously (*Leuenberger et al., 2009*).Briefly, the significant enrichment of underlying KEGG, GO, and Reactome curated pathways was determined from the hypergeometric distribution and corrected for multiple comparisons using Broad Institute Molecular Signatures Database v4.0 (http://www.broadinstitute.org/gsea/msigdb/). Gene sets with a false discovery rate p-value<0.05 were considered significant. Genes that were up- or down-regulated were identified by selecting genes with logarithmic fold-change ratio ($PPAR\alpha^{-/-}/PPAR\alpha^{+/+}$) > 1.3 (up-regulated genes) or < −1.3 (down-regulated genes). The raw data have been deposited in NCBI Gene Expression Omnibus and made accessible through the GEO database (accession number: GSE39669 and GSE39670).

## Cell sorting

E19.5 fetal hepatic cells were isolated as described above and incubated with PE- or FITC-conjugated monoclonal antibodies against DLK (LS-C179444) (LifeSpan Biosciences, Seatlle, WA) or CK18 (ab52459) (Abcam, Cambridge, MA) at 1:200 dilution in PBS supplemented with 0.1% fetal calf serum according to a protocol previously described (*Tanimizu et al., 2003*). The samples were then

washed with PBS and mixed with 1 µg/ml propidium iodide before cell sorting using a FACSAria cell sorter (Becton Dickinson, San Jose, CA).

## Flow cytometry

Primary hepatocytes were isolated as described above. Mitochondria and peroxisomes were stained using MitoTracker Red and SelectFX Alexa Fluor 488 Peroxisome Labeling Kits (Life Technologies, Carlsbad, CA) according to the manufacturer's protocols. Negative controls without Mitotracker Red or fluorophore-conjugated secondary antibodies were used to gate the quadrants. Levels of mitochondrial membrane potential and intracellular ROS production were determined by tetramethylrhodamine, ethyl ester (TMRE) (ab113852) staining and 2',7' –dichlorofluorescin diacetate (DCFDA) (ab113851) (Abcam, Cambridge, MA) staining, respectively, according to the manufacturer's protocols. Treatments with carbonyl cyanide 4-(trifluoromethoxy)phenylhydrazone (FCCP, an ionophore uncoupler of oxidative phosphorylation capable of eliminating mitochondrial membrane potential) at 20 µM for 10 min or N-acetylcysteine (NAC, an antioxidant) at 5 mM for 2 hr served as negative controls for the respective staining. A total of 10,000 events were recorded. The data were analyzed using BD FACSDiva software (version 6; Becton Dickinson, San Jose, CA) and further processed by FlowJo software (version 7.6.1; Tree Star, OR).

## Measurement of Cellular ATP Production

Primary hepatocytes were isolated and cultured as described above. Cellular ATP production was measured in the presence of 25 mM glucose, 10 mM galactose, or 2.5 µM rotenone (Sigma-Aldrich, Saint Louis, MO) using the ENLITEN ATP Assay System Bioluminescence Detection Kit (Promega, Madison, MI) according to the manufacturer's protocol. Bioluminescence signals were read on a Glo-Max 20/20 Luminometer (Promega, Madison, MI) and normalized to the total number of viable cells. Cell viability assay was performed in extracted primary hepatocytes using trypan blue exclusion test based on a protocol previously described (*Strober, 2001*). The production of pyruvate via the glycolytic metabolism of glucose yields 2 net ATP, but the same catalytic pathway yields no net ATP when galactose is used instead of glucose, thereby forcing cells to rely on oxidative phosphorylation for energy. Thus, the use of galactose in the primary hepatocyte culture acts as a positive control for the measurement of oxidative phosphorylation-dependent energy production (*Aguer et al., 2011*).

## Oxygen Consumption Rate (OCR) Measurement

Primary hepatocytes were isolated as described above and $2 \times 10^4$ cells seeded per well on XF-24 cell culture plates (Seahorse Bioscience, Billerica, MA) coated with rat tail collagen in DMEM medium containing 5% FCS and incubated overnight at 37°C. The cellular oxygen consumption rate was measured as described previously (*Nasrin et al., 2010*). The next day, cells were equilibrated with buffer (111 mM NaCl, 4.7 mM KCl, 2 mM $MgSO_4$, 1.2 mM $Na_2HPO_4$, 2.5 mM glucose, 0.5 mM carnitine) and incubated at 37°C for 60 min. The basal cellular respiration was measured without any treatment for 15 min (n = 9 per group), followed by treatment with 200 µM palmitate conjugated with BSA in a 6:1 molar ratio or BSA alone for 40 min (n = 6 per group). To measure the mitochondrial fatty acid oxidation, cells were subsequently incubated with 300 µm etomoxir, a known carnitine palmitoyltransferase I inhibitor, for another 30 min (n = 3 per group). All measurements utilized the Seahorse XF24 Flux Analyzer (Seahorse Biosciences, Billerica, MA). The OCR of whole hepatocytes was standardized for total protein concentration after the assay was completed.

## Histological analysis

Fresh liver samples were collected and embedded in OCT tissue freezing medium (Leica Microsystems, Wetzlar, Germany). Oil Red O stock solution was prepared by dissolving 0.5 g of Oil Red O powder (Sigma-Aldrich, St. Louis, MO) in 500 ml isopropanol. To constitute 50 ml of 60% Oil Red O working solution, 30 ml of the stock solution was diluted with 20 ml of water. Fresh-frozen samples were sectioned (6 µm thick) and stained with 60% Oil Red O solution for 10 min. The Oil Red O-stained sections were counterstained with methylene blue.

**Table 4.** Control diet formulation.

**Manufacturer:** KLIBA NAFAG, SWITZERLAND
**Product ID:** 2222

| Mouse and rat | Experimental diet, purified diet |
|---|---|
| | AIN-93G |
| **Major nutrients** | |
| Dry matter | 90.0% |
| Crude protein | 18.0% |
| Crude fat | 7.0% |
| Crude fiber | 3.5% |
| Crude ash | 3.0% |
| Nitrogen-free extract (NFE) | 58.5% |
| Gross energy | 17.5 MJ/kg |
| Metabolic energy | 15.9 MJ/kg |
| Starch | 35.0% |
| **Amino acids** | |
| Arginine | 0.65% |
| Lysine | 1.40% |
| Methionine | 0.50% |
| Methionine + cystine | 0.85% |
| Tryptophan | 0.22% |
| Threonine | 0.70% |
| **Major mineral elements** | |
| Calcium | 0.52% |
| Phosphorus | 0.32% |
| Magnesium | 0.08% |
| Sodium | 0.22% |
| Potassium | 0.36% |
| Chlorine | 0.15% |
| **Trace elements** | |
| Iron | 65 mg/kg |
| Zinc | 45 mg/kg |
| Copper | 6 mg/kg |
| Iodine | 0.6 mg/kg |
| Manganese | 12 mg/kg |
| Selenium | 0.2 mg/kg |
| **Vitamins** | |
| Vitamin A | 4000 IU/kg |
| Vitamin $D_3$ | 1000 IU/kg |
| Vitamin E | 100 mg/kg |
| Vitamin $K_3$ | 4 mg/kg |
| Vitamin $B_1$ | 6 mg/kg |
| Vitamin $B_2$ | 6 mg/kg |
| Vitamin $B_6$ | 7 mg/kg |
| Vitamin $B_{12}$ | 0.05 mg/kg |
| Nicotinic acid | 30 mg/kg |

*Table 4 continued on next page*

| | |
|---|---|
| Pantothenic acid | 16 mg/kg |
| Folic acid | 2 mg/kg |
| Biotin | 0.2 mg/kg |
| Choline | 1200 mg/kg |
| Ingredients | |
| Corn starch, casein, dextrose, sucrose, refined soybean oil, cellulose, minerals, vitamins, amino acids | |
| Remarks | |
| - Experimental diet for mice and rats | |
| - Given values are calculated averages in air-dry feed | |
| - Production on demand | |
| Delivery form | |
| Pellets 10 mm round | |
| 2222.PH.A05: | |
| 5 kg in welded aluminium bag 2222.MA.A05: 5 kg in welded aluminium bag | |
| KLIBA NAFAG \| PROVIMI KLIBA AG \| CH-4303 Kaiseraugst \| Tel. +41 61 816 16 16 \| Fax. +41 61 816 18 00 \| kliba-nafag@provimi-kliba.ch \| www.kliba-nafag.ch | |

## Analysis of serum and tissue metabolites

Serum glucose and triglyceride levels were measured using the Accutrend Plus meter (Roche Diagnostics, Indianapolis, IN). Serum β-hydroxybutyrate and liver ALT levels were measured using the β-hydroxybutyrate Assay Kit and ALT Activity Assay Kit, respectively (Sigma-Aldrich, Saint Louis, MO). Lipids were extracted from the liver samples by the Bligh/Dyer method and analyzed by gas-liquid chromatography as previously described (*Zadravec et al., 2010*).

## High-fat-diet weaning challenge

Although feeding the P5 pups with milk harvested from *Ppara*[+/-] dams would be ideal for this HFD-weaning challenge, we were ethically and technically restricted to perform oral gavage in these young pups and to collect sufficient milk for this experiment. For these reasons, we resorted to the best compromise by exposing the pups to a high-fat/low-carbohydrate (HF/LC) diet or a control diet from P5 (i.e., just before tooth eruption) to P15 (i.e., when liver steatosis is mostly resolved). The diets (Provimi Kliba, Kaiseraugst, Switzerland) had the following macronutrient compositions (% w/w: fat, carbohydrates, protein): control, 16.7/64.3/19.0 (#2222) (*Table 4*) and HF/LC, 74.4/6.6/19.0 (#2201) (*Table 5*). Both diets used identical macronutrient sources (fat source: beef tallow; carbohydrate source: starch; protein source: casein), allowing a precise comparison of differences in macronutrients.

## L-cycloserine treatment

Two-day-old pups were injected with L-cycloserine (Sigma-Aldrich, Saint Louis, MO) at 30 mg/kg/day or 0.9% saline (vehicle control) for 2 to 4 days. Blood glucose and body weight of the pups was monitored before and after treatment.

## Western blot analysis

Cytoplasmic and nuclear protein fractions were isolated as described above. For Western blotting, equal amounts of protein extracts (20 μg) were resolved by sodium dodecyl sulfate-polyacrylamide gel electrophoresis and electrotransferred onto a nitrocellulose membrane. The membranes were processed as recommended by the antibody suppliers. Chemiluminescence was detected using the Luminata Crescendo Western HRP Substrate (Millipore, MA). Beta-tubulin and U2AF65 were used to check for equal loading and the transfer of cytoplasmic and nuclear proteins, respectively. Primary

**Table 5.** High-fat diet formulation.

**Manufacturer**: KLIBA NAFAG, SWITZERLAND
**Product ID**: 2201

| Mouse and rat | Experimental diet, purified diet |
|---|---|
| Ketogenic diet XL75:XP10 | |
| **Major nutrients** | |
| Dry matter | 99.1% |
| Crude protein | 9.9% |
| Crude fat | 74.4% |
| Crude fiber | 5.5% |
| Crude ash | 6.3% |
| Nitrogen-free extract (NFE) | 3.0% |
| Metabolic energy | 7208 kcal/kg |
| Starch | 0.7% |
| **Amino acids** | |
| Arginine | 0.35% |
| Lysine | 0.79% |
| Methionine | 0.28% |
| Methionine + cystine | 0.90% |
| Tryptophan | 0.13% |
| Threonine | 0.38% |
| **Major mineral elements** | |
| Calcium | 0.98% |
| Phosphorus | 0.61% |
| Magnesium | 0.15% |
| Sodium | 0.40% |
| Potassium | 0.69% |
| Chlorine | 0.57% |
| **Trace elements** | |
| Iron | 151 mg/kg |
| Zinc | 97 mg/kg |
| Copper | 16 mg/kg |
| Iodine | 1.4 mg/kg |
| Manganese | 31 mg/kg |
| Selenium | 0.6 mg/kg |
| **Vitamins** | |
| Vitamin A | 8000 IU/kg |
| Vitamin $D_3$ | 2000 IU/kg |
| Vitamin E | 200 mg/kg |
| Vitamin $K_3$ | 9 mg/kg |
| Vitamin $B_1$ | 12 mg/kg |
| Vitamin $B_2$ | 13 mg/kg |
| Vitamin $B_6$ | 14 mg/kg |
| Vitamin $B_{12}$ | 0.1 mg/kg |
| Nicotinic acid | 66 mg/kg |
| Pantothenic acid | 32 mg/kg |
| Folic acid | 5 mg/kg |

*Table 5 continued on next page*

*Table 5 continued*

**Manufacturer**: KLIBA NAFAG, SWITZERLAND
**Product ID**: 2201

| Mouse and rat | Experimental diet, purified diet |
|---|---|
| Ketogenic diet XL75:XP10 | |
| Biotin | 0.4 mg/kg |
| Choline | 1975 mg/kg |
| Ingredients | |
| Beef fat, casein, cellulose, minerals, vitamins, amino acids | |
| Remarks | |
| - Experimental diet for mice and rats | |
| - Given values are calculated averages in air-dry feed | |
| - Production on demand | |
| Delivery form | |
| Paste | |
| 2201.MA.A05: | |
| 5 kg in welded aluminium bag | |
| KLIBA NAFAG \| PROVIMI KLIBA AG \| CH-4303 Kaiseraugst \| Tel. +41 61 816 16 16 \| Fax. +41 61 816 18 00 \| kliba-nafag@provimi-kliba.ch \| www.kliba-nafag.ch | |

antibody against full-length ANGPTL4 was a kind gift from Prof. Andrew Tan Nguan Soon. Primary antibodies against ACOX1 (#sc-98499), FGF21(#sc-292879), CYP4A14 (#sc-46087), β-tubulin (#sc-9104), U2AF65 (#sc-48804), GR (#sc-1004), and HDAC3 (#sc-17795), and all secondary antibodies conjugated with horseradish peroxidase were acquired from Santa Cruz Biotechnology (Dallas, TX). PPARα polyclonal antibody (#101710) was purchased from Cayman Chemical (Ann Arbor, MI).

## Ex vivo tissue culture

Fetal livers (E19.5) were sliced < 0.1-mm-thick using two adjacent scalpel blades and cultured on collagen-coated plates at 37°C for 24 hr in DMEM:F12 with 10% fetal bovine serum containing 5 mM butyrate, 5 mM β-hydroxybutyrate, or 0.3 μM trichostatin A. All of these chemicals were acquired from Sigma-Aldrich (Saint Louis, MO).

## statistical analysis

We performed power analysis for sample size estimation using Power And Precision software (version 4) by Biostat Inc. when the study was being designed. We set the mean outcome for $Ppara^{+/+}$ and $Ppara^{-/-}$ mice at 100 and 80, respectively (i.e., 20% difference) and estimated the standard deviation to be ± 10% based on our preliminary data. We estimated that a total of 6 mice per group is needed in order to have a power of 80%, which means there is an 80% likelihood that the study will yield a statistically significant (i.e., α value = 0.05) effect to allow us to detect a mean outcome difference of 20% between $Ppara^{+/+}$ and $Ppara^{-/-}$ mice. The outcomes (e.g., gene expression, metabolic parameters) are measured on a continuous scale. The null hypothesis is that the mean outcome for these two groups is identical. The computation of the sample size is based on the assumption that there would be no missing data (i.e., all mice will produce data). The data in all figure panels reflect multiple experiments performed on different days using pups (n = 6 with triplicates) derived from different litters. In this study, we used 6 mice per group (i.e., biological replicates) for all experiments. Further, measurements and experiments were repeated in triplicates for each sample (i.e., technical replication). Values were expressed as mean ± standard error of the mean (SEM). Statistical tests, including the two-tailed Mann-Whitney and two-way ANOVA with Bonferroni post-hoc analysis, were performed using GraphPad Prism software (version 5.00). p-values<0.05 were considered significant.

## Acknowledgements

We thank Béatrice Desvergne, Federica Gilardi, Sven Pettersson, and Andrew Nguan Soon Tan for discussions; Michaël Baruchet, Corinne Tallichet-Blanc, and Mélanie Schuh for technical assistance; and the Mouse Clinic Institute in Illkirch, France, for the gift of *Nr3c1*$^{-/-}$ mice. We also thank Pierre Chambon and Daniel Metzger for providing GR-mutated liver samples. Gene expression profiling was performed in collaboration with the Lausanne Genomic Technology Facility. We thank Nesibe Peker from the Singapore Institute for Clinical Sciences for their technical assistance in measuring the oxygen consumption rate.

## Additional information

### Funding

| Funder | Grant reference number | Author |
|---|---|---|
| Schweizerischer Nationalfonds zur Förderung der Wissenschaftlichen Forschung | | Walter Wahli |
| Nanyang Technological University | Lee Kong Chian School of Medicine Start-Up Grant | Walter Wahli |
| Bonizzi-Theler Stiftung | | Walter Wahli |
| Human Frontier Science Program | | Walter Wahli |
| the 7th EU Program TORNADO | | Walter Wahli |
| Université de Lausanne | | Walter Wahli |
| Pôle de Recherche National "Frontiers in Genetics" | | Walter Wahli |

The funders had no role in study design, data collection and interpretation, or the decision to submit the work for publication.

### Author contributions

GR, CKT, Conception and design, Acquisition of data, Analysis and interpretation of data, Drafting or revising the article ; NK, Acquisition of data, Analysis and interpretation of data; AM, NL, JB-M, EP, HG, Contributed to the experiments, Data acquisition, Revised the manuscript, Acquisition of data, Analysis and interpretation of data; WW, Designed the study and was involved in all aspects of the experiments, Wrote the manuscript, Supervised this study, Analysis and interpretation of data

### Author ORCIDs

Walter Wahli, http://orcid.org/0000-0002-5966-9089

### Ethics

Animal experimentation: This study was performed in strict accordance with the recommendations in the Guide for the Care and Use of Laboratory Animals of the National Institutes of Health. All of the animals were handled according to the institutional animal care and use committee (IACUC) protocol (#2013/SHS/866) approved by SingHealth, Singapore and the Vaud Cantonal Authority, Switzerland.

## Additional files

### Major datasets

The following datasets were generated:

| Author(s) | Year | Dataset title | Dataset URL | Database, license, and accessibility information |
|---|---|---|---|---|
| Gianpaolo R, Nourhène K, Walter W | 2016 | Prenatal PPARa-dependent gene expression in fetal mouse liver just before birth (E19.5) | http://www.ncbi.nlm.nih.gov/geo/query/acc.cgi?acc=GSE39669 | Publicly available at the NCBI Gene Expression Omnibus (Accession no: GSE39669) |
| Gianpaolo R, Nourhène K, Walter W | 2016 | Postnatal PPARa-dependent gene expression in two-days old mouse liver | http://www.ncbi.nlm.nih.gov/geo/query/acc.cgi?acc=GSE39670 | Publicly available at the NCBI Gene Expression Omnibus (Accession no: GSE39670) |

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
