## [Decision Letter]

Thank you for submitting your work entitled "Prenatal glucocorticoid signaling is required for PPARα activity and epigenetic regulation of FGF21 at birth" for consideration by *eLife*. Your article has been reviewed by three peer reviewers, and the evaluation has been overseen by K VijayRaghavan as the Senior Editor who also served as Reviewing Editor. The following individual involved in review of your submission has agreed to reveal their identity: Sander Kersten (peer reviewer).

The reviewers have discussed the reviews with one another and the Reviewing Editor has drafted this decision to help you prepare a revised submission.

Summary:

In this paper Rando, G et al. elegantly show a new mechanism by which PPARα controls hepatic adaptation to changes in the nutrition during the late stages of development and at birth. The story is very interesting as general knowledge and it adds a novel model by which the liver may adapt to changes in the diet and how this adaptation could be exploited in health and disease. Specifically, the paper explores the mechanism underlying the activation of the transcription factor PPARα in liver during the late embryonic period and the impact of this mechanism on cellular energetics and lipid metabolism. The paper highlights the crucial role of the glucocorticoid receptor in activation of PPARα during embryogenesis, and proposes a contribution of epigenetic mechanisms involving the ketone body β-hydroxybutyrate. The paper consists of a number of mini-stories that differ substantially in the quality of the data and in the level of depth provided. Some of the mini-stories are mostly confirmatory, while others are novel. The unification of the individual stories into a coherent and convincing overall message is a bit problematic and deserves attention. Even though the experiments are very well conducted, there are some pitfalls that should be resolved for a better understanding of the main points of the story.

The finding that PPARα target genes exhibit different types of expression profiles pre- and post-natally is worthy of a more detailed mechanistic pursuit. The demonstration that the glucocorticoid receptor is the driving force behind induction of PPARα during embryogenesis and directly binds to the PPARα promoter is very interesting and novel. The data showing a major impact of PPARα deletion on neonatal lipid metabolism are confirmatory of a publication by Cotter et al. (Am J Physiol Endocrinol Metab. 2014 Jul 15; 307(2): E176-E187). The data on epigenetic regulation of PPARα target genes are interesting but do not link particularly well to the data on lipid metabolism. Β-hydroxybutyrate may contribute to induction of specific PPARα target genes (*Fgf21*) via epigenetic mechanism but how does this effect connect to the observed phenotype of *Ppara^-/-^* mice. The part describing epigenetic regulation by β-hydroxybutyrate is potentially very exciting (following the work of Verdin) but the physiological/metabolic rationale needs to be better explained.

Overall, the paper has quite a few rough edges. The presentation/description of the data and story lines can be much improved. The paper would benefit from a more linear story line from GR to PPARα to ketone bodies to epigenetic regulation of *Fgf21* (while perhaps mentioning a potential vicious cycle between PPARα, ketone bodies and *Fgf21*). Some figures should be removed as they are unnecessary and distracting (confirmatory data of Cotter, data on gluconeogenesis and anaplerotic oxidation of amino acids). It would be fantastic if the role of β-hydroxybutyrate in epigenetic gene regulation of *Fgf21* could be confirmed via in vivo data but that would be an unreasonable request.

All three reviewers mention that the storyline of the paper is complex and that the structure of the paper needs to be substantially improved. That is something that could be completed in two months. As far as additional analyses are concerned, the possible translation of the data to humans should addressed with more caution and in any case be very difficult to address experimentally. We would appreciate additional elaboration on the generalizability of effect of β-hydroxybutyrate on gene expression. The authors are encouraged to look at additional genes to solidify this very interesting finding.

Essential revisions:

We list the major comments from each reviewer rather than consolidating them. They are largely distinct with some overlap.

Reviewer 1:

1) It is described that PPARα null and WT animals were maintained on a mixed C57BL/6 × SV129 background and bred by intercrossing mice heterozygous for the mutation. The rationale for this practice is unclear. First, PPARα null mice are available at Jackson labs on a pure C57BL/6 or SV129 background, which would be preferred over a mixed genetic background. In addition, PPARα null mice are perfectly viable as homozygous. Why weren't PPARα null maintained as homozygous and compared with wildtype mice matched for genetic background?

2) For Figure 1 to be meaningful, the entire set of PPARα targets within a metabolic pathway should be shown (as opposed to a potentially biased subselection of genes) or the comparative analysis should be performed statistically at the pathway level.

3) In the second paragraph of the Results section: the nearly 90% reduction in ATP production in *Ppara^-/-^*hepatocytes as compared to WT hepatocytes seems implausible, especially considering that the cells were fed with glucose or galactose. Could it be that these results are reflective of the low viability of the *Ppara^-/-^* hepatocytes? There is no mentioning of measurement of cell viability in the primary hepatocytes.

4) Please explain why palmitate and etomoxir do not have any influence on oxygen consumption in either *Ppara*^+/+^ or *Ppara^-/-^* hepatocytes. The lack of effect of etomoxir on oxygen consumption in *Ppara*^+/+^ hepatocytes suggests that the difference in oxygen consumption between *Ppara*^+/+^ and *Ppara^-/-^*hepatocytes is not related to differences in fatty acid oxidation.

5) The part of the manuscript addressing plasma glucose (subsection “PPARα Required for Protection against Neonatal Hepatic Steatosis”, last paragraph) is poorly developed. It is unclear why a *lack* of difference in plasma glucose levels or body weight between the two genotypes would prompt the authors to check whether the gluconeogenesis pathway is modified.

6) It is described that "… increased anaplerotic oxidation of amino acids may be an alternative pathway for maintaining the de novo synthesis of glucose as an energy source when the FAO capacity is impeded in the absence of PPARα". What is the evidence that oxidation of amino acids may be increased? The genes shown in Figure 3—figure supplement 2E are part of an energy-requiring process: gluconeogenesis.

7) In the Discussion section, increased anaplerotic oxidation of amino acids is proposed as a potential explanation for the discrepancy with Cotter et al., who found that *Ppara^-/-^* pups were hypoglycemic. Why would anaplerotic oxidation of amino acids be different between the two studies?

8) The interpretation of the experiment with L-cycloserine is difficult. Did the authors verify that it effectively inhibited gluconeogenesis? For instance, were plasma glucose levels altered. The authors don't derive any conclusion from this experiment. Why was it included?

9) In the last paragraph of the Results section: Rephrase "Activation of the *Fgf21* TSS and the levels of *Fgf21* mRNA required both the activation of PPARα by Wy and HDAC3 inhibition through butyrate or β-hydroxybutyrate treatment." Can the authors explain why WY has no effect on *Fgf21* mRNA or transcriptional activation in the absence of butyrate of β-hydroxybutyrate (Figure 7), whereas WY alone effectively induces polII binding to the *Fgf21* TSS in Figure 5.

Reviewer 2:

1) Hepatocytes from E14.5 and E15.5 did not show any changes in the activity of *Pparα*, which is not surprising since at this stage of liver development the liver bud generates mainly hematopoietic cells and precursors. At later stages we can expect that some hepatic precursors appear and at 19.5 there is a mixed population of cells in the liver: one compartment is the undifferentiated hepatoblasts and the other is the newly formed hepatocytes. Since the deletion of *Pparα* is constitutive, it would be important to show if either the changes seen at E19.5 are stemming from the pluripotent hepatoblasts compartment or from the hepatocytes. The same is observed in the newly born population. This is a key point to be addressed since it could change the main story. It is expected that hepatoblasts will show an aerobic glycolytic metabolic profile, as seen in many other stem cell-like population; but it would be important to address what happens to hepatocytes before birth?

2) In the *Ppara^-/-^*mice there are many changes but the most relevant one is related to the number of mitochondria. How can it be excluded that all the changes in FAO and gluconeogenesis observed are not cellular-autonomous metabolic adaptation as a result of the lower number of mitochondria instead of epigenetic regulation? Is the mitochondrial potential maintained in these mitochondria? What is the redox status of the mitochondria and the cytoplasm? Can it be excluded that the induction of β-Hydroxybutyrate is not due to mitochondrial defects?

3) Related to the previous question, the authors claim that ALT could be activated to replenish the TCA cycle in the absence of *Pparα*. This is very interesting, but not backed up by solid data. The ALT activity is mainly cytoplasmic because the liver expresses mainly the cytoplasmic isoform of Gpt (ALT). Therefore, if the reactions occur in the cytoplasm, it is expected to be accompanied by changes in the mitochondrial potential. This would need to be shown. In some cases, the change in the pyruvate flux to alanine will affect amino acid metabolism. Were any changes in the expression of genes from amino acid metabolism detected in the microarray data? In addition, is there a reason that ALT would be more active?

4) For the ChIP-seq data, the authors should show that the antibody against GR does not pull down anything in the *GR*^-/-^ livers.

5) The paper is written in a complicated fashion, which makes understanding of the most important points tedious. The authors should be encouraged to simplify their writing.

Reviewer 3:

1) The mice used are classical PPARα k.o. mice. How specific are the effects found for the liver (in particular in vivo) as the PPARα is expressed in many organs (including the gut) and this could influence the outcome?

2) The hepatic steatosis in *Ppara^-/-^* is caused by? The dietary fat or more related to the carb part of the diet leading to de novo lipogenesis that is also under control of PPARα, I thought (references are not mentioned describing the role in regulating 'new' fat metabolism). Translated to the human situation early breastmilk has a rather low fat content and as a baby breastfeeds, the fat content very gradually increases, with the milk becoming fattier and fattier over time. So in the beginning is largely a sweet high lactose/carb feed; how does this relate to this study?

3) Related to this the HF diets used for the challenge are quite extreme, ketogenic diets, why this is needed as I guess this is not the same in the human situation? I do not see the full relevance for this challenge? To induce FGF21? Is FGF21 functional for early nutrition related metabolic regulation or more related to starvation/low calorie periods of the baby (where β-hydroxybutyrate also goes up?), so is it ketogenesis because of ketogenic diets or fasting-related metabolic states? I am not really convinced of the diets used here to mimic (human) milk. Human milk is clearly not a ketogenic diet. So I question the regulation of FGF21 because of the milk.

4) The composition is not precisely mentioned e.g. no lactose or sucrose in the diet?). On the other hand, using chow is not ideal as it will prevent the exact reproducibility of this study because of the natural/variable ingredients of the chow.

In summary, I am not so convinced yet that the main finding of this paper (epigenetic regulation of FGF21, which has been partly described before, and partly attributed to SIRT1/3) is really related to the HF diet challenge but it could be also alternatively explained as it might contribute in the difficult periods of low-/no-food /starvation when bHB is providing fuel for the brain to optimize glucose transport by other organs to prevent hypoglycemia etc.?

---

## [Author Response]

Summary:

[…] All three reviewers mention that the storyline of the paper is complex and that the structure of the paper needs to be substantially improved. That is something that could be completed in two months. As far as additional analyses are concerned, the possible translation of the data to humans should addressed with more caution and in any case be very difficult to address experimentally. We would appreciate additional elaboration on the generalizability of effect of β-hydroxybutyrate on gene expression. The authors are encouraged to look at additional genes to solidify this very interesting finding.

Thank you for this comprehensive summary, which highlights the strengths and weaknesses of our manuscript. We have addressed all the reviewers’ comments in the following point-by-point response.

In response to some of the comments highlighted in this summary but not raised by the reviewers, we have made the following changes in the manuscript:

We have changed the title of the manuscript to “Glucocorticoid receptor-PPARα axis in fetal mouse liver prepares neonates for milk lipid catabolism” to better encapsulate the major findings reported in our revised manuscript.

We have reconstructed the manuscript to streamline the story (from GR to PPARα to ketone bodies to epigenetic regulation of *Fgf21*) and simplified our writing to enhance accessibility to a broad readership. We have also discussed and placed more emphasis on the physiological relevance of our work in the Discussion.

We have now established the connection between epigenetic regulation of PPARα target genes (particularly FGF21) and lipid metabolism in the Discussion (sixth paragraph). In particular, the physiological/metabolic relevance of the epigenetic regulation of FGF21 by β-hydroxybutyrate has been discussed.

We have also established the connection between the induction of FGF21 via HDAC3 inhibition by β-hydroxybutyrate and the observed phenotype of *Ppara^-/-^*mice in the Discussion (sixth paragraph).

We did not remove some of the figures as suggested (e.g., gluconeogenesis and anaplerotic oxidation of amino acids) because we believe that these data complement the work of Cotter et al. and provide important information on the overall metabolic situations in the absence of PPARα. We believe that you found the inclusion of these data unnecessary and distracting because of our failure to successfully integrate/connect this information with other parts of the metabolic processes. We have tried to address this shortfall in the Discussion of the revised manuscript (fourth paragraph).

We have mentioned, as suggested, a potential vicious cycle between PPARα, ketone bodies and FGF21 in the Discussion (last paragraph). We have also constructed a schematic illustration of this vicious cycle (Figure 8).

Due to time constraint for resubmission, we did not validate the role of β-hydroxybutyrate in epigenetic regulation of *Fgf21* in vivo. However, we have further discussed the work by Verdin and coworkers (Shimazu et al., 2013) (PMID: 23223453) in which β-hydroxybutyrate was identified in vivo as the endogenous HDAC inhibitor (Discussion, fifth paragraph).

We have addressed the possible translation of the data to human situations in the Discussion with more caution (sixth paragraph).

We have included additional data on another PPARα regulated gene, *Angptl4*, which like *Fgf21* functions as an adaptive regulator of metabolism whose expression is also governed by the β-hydroxybutyrate-mediated HDAC3 inhibition. The new data can be found in Figure 6. It is important to note that, unlike *Fgf21*, the expression of *Angptl4* started before birth. We have reported this new data in the Results (subsection “An Epigenetic Switch Controls a Subset of PPARα Target Genes”, last paragraph) and further discussed this difference in the Discussion (sixth paragraph).

Further, we have also performed several additional experiments and we believe that the new data have strengthened our work. We hope that the revised version of the manuscript will meet your expectations. Thank you very much for your consideration and the constructive comments that have helped us improve our manuscript.

Essential revisions:

*We list the major comments from each reviewer rather than consolidating them. They are largely distinct with some overlap.*

Reviewer 1:

1) It is described that PPARα null and WT animals were maintained on a mixed C57BL/6 × SV129 background and bred by intercrossing mice heterozygous for the mutation. The rationale for this practice is unclear. First, PPARα null mice are available at Jackson labs on a pure C57BL/6 or SV129 background, which would be preferred over a mixed genetic background. In addition, PPARα null mice are perfectly viable as homozygous. Why weren't PPARα null maintained as homozygous and compared with wildtype mice matched for genetic background?

We thank the reviewer for bringing to our attention this lack of clarity in the rationale of our breeding strategy. In addition, this good comment made us realize that there was a mistake in our text.

Firstly, we acquired PPARα-knockout (*Ppara^-/-^*) mice on a pure C57BL/6 background from the Jackson Laboratory and inter-crossed them with our in-house wild-type (WT) C57BL/6 mice for at least 5 generations to generate PPARα-heterozygous (HT) mice. We have corrected this information in the Materials and methods (first paragraph).

Secondly, we obtained the foetuses and pups through heterozygous mating so that we could control for gestational and lactational confounding factors introduced by using dams from different genotypes (WT and *Ppara^-/-^*). It is currently unknown whether PPARα deficiency in lactating dams can lead to alteration in milk composition, as it is the case for PPARγ-knockout mice (Wan et al., 2007) (PMID: 17652179). Therefore, it was important to ensure that both WT and *Ppara^-/-^*pups received milk of the same nutritional content from the HT dams. We have clarified this rationale in the Materials and methods (subsection “ChIP Assays”).

2) For Figure 1 to be meaningful, the entire set of PPARα targets within a metabolic pathway should be shown (as opposed to a potentially biased subselection of genes) or the comparative analysis should be performed statistically at the pathway level.

We have performed an unbiased comparative analysis of the similarly and differentially regulated (both up- and down-regulation) genes between WT and *Ppara^-/-^*foetuses at E19.5 and pups P2. Statistical analysis was also performed with *p-*values indicated for the top 10 most affected pathways for each category. This piece of information can be found in [Supplementary-material SD1-data]).

3) In the second paragraph of the Results section: the nearly 90% reduction in ATP production in Ppara^-/-^ hepatocytes as compared to WT hepatocytes seems implausible, especially considering that the cells were fed with glucose or galactose. Could it be that these results are reflective of the low viability of the Ppara^-/-^ hepatocytes? There is no mentioning of measurement of cell viability in the primary hepatocytes.

We thank the reviewer for this comment. As suggested, we have repeated the ATP measurements using primary hepatocytes obtained from E19.5 and P2 livers (n=4 per time point per group). We have performed trypan blue exclusion test to quantify the number and percentage of viable cells. We observed a ~20% reduction in the number of viable primary *Ppara^-/-^*hepatocytes as compared with the WT hepatocytes at both time points, despite that these two groups were simultaneously extracted using the same procedure (see Figure 9, left panel). We subsequently normalized ATP production to the number of viable cells and found that ATP production in *Ppara^-/-^*hepatocytes was ~40-50% of the levels measured in both prenatal and postnatal WT controls (Figure 3 or Figure 9, right panel). We have reported the new data in the Results (subsection “PPARα Controls the Prenatal Lipid Catabolic Machiner”, last paragraph).

and Ppara^-/-^ hepatocytes is not related to differences in fatty acid oxidation.

Author response image 1.**DOI:**
http://dx.doi.org/10.7554/eLife.11853.020

*4) Please explain why palmitate and etomoxir do not have any influence on oxygen consumption in either Ppara*^+/+^
*or Ppara^-/-^ hepatocytes. The lack of effect of etomoxir on oxygen consumption in Ppara*^+/+^
*hepatocytes suggests that the difference in oxygen consumption between Ppara*^+/+^*and Ppara^-/-^ hepatocytes is notrelated to differences in fatty acid oxidation.*

We apologise for the confusion caused as we previously did not indicate the statistical significance between the treatment groups of the same genotype. We have now performed two-tailed Mann-Whitney *U* test for these observed differences in palmitate treatment groups (where ^###^ indicates *P* < 0.001 vs. baseline), and in palmitate + etomoxir treatment groups (where ^§§§^ indicates *P* < 0.001 vs. palmitate treatment alone). We found that the differences were statistically significant in WT but not in *Ppara^-/-^* groups. To clarify this, we have included the indications of significance in Figure 3.

5) The part of the manuscript addressing plasma glucose (subsection “PPARα Required for Protection against Neonatal Hepatic Steatosis”, last paragraph) is poorly developed. It is unclear why a lack of difference in plasma glucose levels or body weight between the two genotypes would prompt the authors to check whether the gluconeogenesis pathway is modified.

We thank the reviewer for highlighting this lack of clarity in the rationale of investigating the gluconeogenesis pathway. It is known that the transition from a carbohydrate-rich fetal diet to a high-fat, low-carbohydrate neonatal diet requires inductions of hepatic fatty acid oxidation (FAO), gluconeogenesis, and ketogenesis in order to preserve bioenergetics homeostasis (Cotter et al., 2013) (PMID: 23689508). In addition, an effective hepatic FAO is necessary to support hepatic de novo ketogenesis and gluconeogenesis by providing essential co-factors such as acetyl-CoA and/or NADH (Girard, 1986) (PMID: 3542066). Since PPARα deficiency leads to defective hepatic FAO and hypoketonemia, we were interested to find out whether the gluconeogenesis pathway is altered to sustain the energy demand for normal growth. We have clarified this point in the Results (subsection “PPARα Is Required for Protection against Neonatal Hepatic Steatosis”, last paragraph).

6) It is described that "… increased anaplerotic oxidation of amino acids may be an alternative pathway for maintaining the de novo synthesis of glucose as an energy source when the FAO capacity is impeded in the absence of PPARα". What is the evidence that oxidation of amino acids may be increased? The genes shown in Figure 3—figure supplement 2E are part of an energy-requiring process: gluconeogenesis.

Based on our microarray data, we have delineated that most of genes coding for the rate-limiting enzymes involved in amino acid oxidation were concomitantly up-regulated in the *Ppara^-/-^* liver at P2 (except for valine, leucine, and isoleucine). Up-regulation of these genes provides some of the intermediate substrates in TCA/Krebs cycle, such as α-ketoglutarate, fumarate, and oxaloacetate, which are ultimately used for gluconeogenesis. We have incorporated this information in the Results (subsection “PPARα Is Required for Protection against Neonatal Hepatic Steatosis”, last paragraph) and in Figure 4—figure supplement 2 with the gene name indicated in the figure legend.

7) In the Discussion section, increased anaplerotic oxidation of amino acids is proposed as a potential explanation for the discrepancy with Cotter et al., who found that Ppara^-/-^ pups were hypoglycemic. Why would anaplerotic oxidation of amino acids be different between the two studies?

We thank the reviewer for highlighting this point. The hypoglycemic phenotype was reported by Cotter et al. in *Ppara^-/-^* pups at P1 due to a decreased hepatic production of glucose from glycerol. However, we did not detect any sign of hypoglycemia in *Ppara^-/-^*neonates starting from P2. We believe the findings from the two groups complement each other. This apparent discrepancy is likely attributable to the rapid (~ 1 day) and dynamic adaptive function of the neonatal liver in maintaining glucose homeostasis via anaplerotic amino acid oxidation, which was particularly evident in the *Ppara^-/-^* liver at P2. We have further discussed this discrepancy and emphasized on this implication in the Discussion (fourth paragraph).

8) The interpretation of the experiment with L-cycloserine is difficult. Did the authors verify that it effectively inhibited gluconeogenesis? For instance, were plasma glucose levels altered. The authors don't derive any conclusion from this experiment. Why was it included?

We thank the reviewer for the comment. We have now measured the plasma glucose levels in WT and *Ppara^-/-^* pups after L-cycloserine treatment. We found that L-cycloserine treatment consistently impeded the production of glucose in *Ppara^-/-^* pups at P2, P4, P5 and P6. This finding further supports the notion that anaplerotic amino acid oxidation in *Ppara^-/-^* liver provides the intermediate substrates in TCA/Krebs cycle required for the gluconeogenesis pathway (Figure 4—figure supplement 2). We have included this information in the Results (subsection “PPARα Is Required for Protection against Neonatal Hepatic Steatosis”, last paragraph) and in Figure 4—figure supplement 2.

9) In the last paragraph of the Results section: Rephrase "Activation of the Fgf21 TSS and the levels of Fgf21 mRNA required both the activation of PPARα by Wy and HDAC3 inhibition through butyrate or β-hydroxybutyrate treatment." Can the authors explain why WY has no effect on Fgf21 mRNA or transcriptional activation in the absence of butyrate of β-hydroxybutyrate (Figure 7), whereas WY alone effectively induces polII binding to the Fgf21 TSS in Figure 5.

We thank the reviewer for highlighting this interesting point. It was previously indicated that HDAC-mediated histone deacetylation may inhibit transcription at the initiation and/or elongation step (Wang et al., 2009) (PMCID: PMC2750862). If the genes are loaded by Pol2 at the TSS, similar to what we observed at the *Fgf21* TSS upon PPARα agonistic activation, HDAC-mediated inhibition of transcription most likely happens at the elongation step although abortive initiation by the TSS bound Pol2 cannot be excluded. Our results suggest that PPARα activation induces the recruitment and assembly of transcription initiation complex at the *Fgf21* TSS. However, the elongation of *Fgf21* mRNA transcript only ensues after alleviation of HDAC3 inhibition by β-hydroxybutyrate. Therefore, it is consistent with our findings that WY-14643 has no effect on *Fgf21* mRNA or transcriptional activation in the absence of butyrate or β-hydroxybutyrate due to HDAC3-mediated inhibition. We have discussed this in the Discussion (fifth paragraph).

We have also rephrased the sentence (Results), which now reads “Activation of the *Fgf21* transcriptional activity was dependent on both the activation of PPARα and HDAC3 inhibition through butyrate or β-hydroxybutyrate treatment”.

Reviewer 2:

1) Hepatocytes from E14.5 and E15.5 did not show any changes in the activity of Pparα, which is not surprising since at this stage of liver development the liver bud generates mainly hematopoietic cells and precursors. At later stages we can expect that some hepatic precursors appear and at 19.5 there is a mixed population of cells in the liver: one compartment is the undifferentiated hepatoblasts and the other is the newly formed hepatocytes. Since the deletion of Pparα is constitutive, it would be important to show if either the changes seen at E19.5 are stemming from the pluripotent hepatoblasts compartment or from the hepatocytes. The same is observed in the newly born population. This is a key point to be addressed since it could change the main story. It is expected that hepatoblasts will show an aerobic glycolytic metabolic profile, as seen in many other stem cell-like population; but it would be important to address what happens to hepatocytes before birth?

We thank the reviewer for the suggestions. Since there is a heterogeneous population of undifferentiated hepatoblasts and differentiated hepatocytes in the liver at E19.5, we determined whether hepatoblasts and/or hepatocytes were responsible for the observed changes in mRNA expression. Using antibodies against hepatoblast- and hepatocyte-specific marker (i.e., DLK and CK18, respectively), we performed flow cytometric analyses on the cells extracted from *Ppara*^+/+^ and *Ppara^-/-^* livers at E19.5. Upon cell sorting, we specifically recovered the DLK^+^ and CK18^+^ fractions. We showed that PPARα deficiency only led to marginal changes in the abundance of these two hepatic cell populations (Figure 2). Furthermore, we determined the mRNA expression of *GR, PPARα* and PPARα target genes in these cells. Our results indicated that both *Ppara*^+/+^ hepatoblasts and hepatocytes similarly contributed to the expression of these genes (Figure 2). Notably, we observed a modest but not significant upward trend in the expression of oxidative genes (e.g., *Acox1, Acaa2, Acadl*, and *Acadvl*) and a downward trend in the expression of glycolytic genes (e.g., *Gck* and *Hk1*) in the *Ppara*^+/+^ hepatocytes compared to hepatoblasts, thereby suggestive of a more pronounced oxidative program in these cells (Figure 2). In contrast, the expression of these genes was concomitantly down-regulated in *Ppara^-/-^* hepatoblasts and hepatocytes, thereby highlighting the pivotal role of PPARα in glycolytic and oxidative metabolism in both cell types. We have reported this new data in the Results (subsection “PPARα Controls the Prenatal Lipid Catabolic Machinery”, second paragraph).

2) In the Ppara^-/-^mice there are many changes but the most relevant one is related to the number of mitochondria. How can it be excluded that all the changes in FAO and gluconeogenesis observed are not cellular-autonomous metabolic adaptation as a result of the lower number of mitochondria instead of epigenetic regulation? Is the mitochondrial potential maintained in these mitochondria? What is the redox status of the mitochondria and the cytoplasm? Can it be excluded that the induction of β-Hydroxybutyrate is not due to mitochondrial defects?

We thank the reviewer for raising this important point. Firstly, many of the PPARα regulated genes involved in FAO have previously been shown to contain PPRE in their promoter region and hence, recognized as direct PPARα target genes (Rakhshandehroo et al., 2010) (PMID: 20936127). This implicates that the reduced oxidative capacity observed in *Ppara^-/-^* mice was due to the lack of direct PPARα-mediated stimulatory effects rather than cellular-autonomous metabolic adaptation in response to lower mitochondria numbers.

Hepatic FAO has previously been shown to increase the gluconeogenic flux by providing acetyl-CoA and NADH which are both necessary for gluconeogenesis (Ferré et al., 1979) (PMID: 508300). This notion is in line with the observation of hypoglycemia in *Ppara^-/-^* pups at P1 (Cotter et al., 2014) (PMID: 24865983). However, we observed normoglycemia in these pups beyond P2. We subsequently delineated that a cell-autonomous adaptation enhanced in the absence of PPARα and mediated through up-regulation of alanine aminotransferase (ALT) and anaplerotic oxidation of amino acid oxidation (as shown in Figure 4—figure supplement 2) was in place to maintain glucose homeostasis so as to sustain normal postnatal growth in *Ppara^-/-^* neonates. Thus, glucose became the primarily source of energy in these mice since they exhibited defective hepatic FAO and ketogenesis. In parallel, we also observed an augmented expression of pyruvate carboxylase (*Pcx)* and other gluconeogenic genes in *Ppara^-/-^* liver (Figure 4—figure supplement 2). This compensatory effect likely corresponds with a lower hepatic acetyl-CoA level, as supported by a recent report that the hepatic acetyl-CoA acts as an allosteric regulator of Pcx (Perry et al., 2015) (PMID: 25662011). We have discussed this information in the Discussion (fourth paragraph).

We have also measured mitochondrial membrane potential (∆Ψm) and intracellular ROS production levels, which are two common parameters used in assessing mitochondria health and functions (Suski et al., 2012) (PMID: 22057568). Interestingly, we observed a much lower mitochondrial membrane potential (∆Ψm) and a markedly increased intracellular ROS level in *Ppara^-/-^* hepatocytes at P2, indicative of mitochondria dysfunction at this stage (Figure 3). However, there was no discernible difference in ∆Ψm between *Pparα*^+/+^ and *Ppara^-/-^* hepatocytes at E19.5 despite that a slight increase in the intracellular ROS level was observed in the latter. These findings suggest that an increased ROS level in cells likely precedes the loss of ∆Ψm and mitochondria dysfunction. This is consistent with a previous report of a similar mitochondrial disorder associated with the dysfunctions of the respiratory chain components (Lebiedzinska et al., 2010) (PMID: 20226758). Particularly, the increased intracellular ROS levels observed in *Ppara^-/-^* hepatocytes may be due to the reduced clearance of ROS as evidenced by the down-regulated of genes coding for mitochondrial superoxide dismutase 2 (SOD2) and copper chaperone for SOD1 before and/or after birth ([Supplementary-material SD1-data]). Particularly, the increased intracellular ROS levels observed in *Ppara^-/-^* hepatocytes may be due to the reduced clearance of ROS as evidenced by the down-regulated of genes coding for mitochondrial superoxide dismutase 2 (SOD2) and copper chaperone for SOD1 before and/or after birth ([Supplementary-material SD1-data]). This observation is consistent with the report that PPARα activation by agonist clofibrate stimulates SOD1 and catalase activities as part of the defence mechanism against oxidative stress in the heart (Ibarra-Lara et al., 2016). We have reported and discussed the new data in the Results (subsection “PPARα Controls the Prenatal Lipid Catabolic Machinery”, last paragraph) and Discussion (third paragraph), respectively.

Lastly, the observed reduction in the serum β-hydroxybutyrate in *Ppara^-/-^* neonates is likely due to defective mitochondrial FAO, which provides the co-factors (e.g., acetyl-CoA and NADH) required for ketogenesis. We have discussed about this point in the Discussion (fourth paragraph).

3) Related to the previous question, the authors claim that ALT could be activated to replenish the TCA cycle in the absence of Pparα. This is very interesting, but not backed up by solid data. The ALT activity is mainly cytoplasmic because the liver expresses mainly the cytoplasmic isoform of Gpt (ALT). Therefore, if the reactions occur in the cytoplasm, it is expected to be accompanied by changes in the mitochondrial potential. This would need to be shown. In some cases, the change in the pyruvate flux to alanine will affect amino acid metabolism. Were any changes in the expression of genes from amino acid metabolism detected in the microarray data? In addition, is there a reason that ALT would be more active?

In agreement with this interesting point raised by the reviewer, most of genes coding for the rate-limiting enzymes involved in amino acid oxidation were concomitantly up-regulated in *Ppara^-/-^* liver at P2 (except for valine, leucine, and isoleucine). Up-regulation of these genes provides some of the substrates in TCA cycle, such as α-ketoglutarate, fumarate, and oxaloacetate, which are ultimately used for gluconeogenesis. We have now incorporated this important information in Figure 4—figure supplement 2. We have also included the gene name of the rate-limiting enzymes involved in the pathways indicated in the figure legend. Thus far, we couldn’t explain the increased *ALT* expression in *Ppara^-/-^* liver except that it involved a pathway up-regulated in the absence of PPARα to compensate for the defective FAO and hypoglycemia. Interestingly, the expression of *ALT2* has previously been shown to be up-regulated under metabolic stress (Salgado et al., 2014, PMID: 24418603). Furthermore, plasma and tissue ALT levels are elevated in response to endoplasmic reticulum (ER) stress (Josekutty et al., 2013) (PMID: 23532846). Interestingly, defective PPARα signaling has been reported to cause hepatic mitochondrial and ER stress in the pathogenesis of hepatic steatosis through reduced mRNA expression of the sarco/endoplasmic reticulum calcium ATPase (SERCA) (Su et al., 2014) (PMID: 24735884). Our microarray data revealed a consistent reduction by ~1.2-fold in the expression of *Atp2a2*, which encodes for SERCA, in *Ppara^-/-^* liver at E19.5 and P2, respectively ([Supplementary-material SD1-data]). We have incorporated this information in the Results (subsection “PPARα Is Required for Protection against Neonatal Hepatic Steatosis”, last paragraph) and in the Discussion (fourth paragraph).

4) For the ChIP-seq data, the authors should show that the antibody against GR does not pull down anything in the GR^-/-^ livers.

We thank the reviewer for the suggestion of this important control. Due to logistic limitation, we were unable to acquire *GR^-/-^*livers to meet the timeline for resubmission. Nonetheless, we have performed siRNA-mediated knockdown of GR in primary hepatocytes extracted from E17 *Pparα^+/+^*livers. Upon treatment with GR-targeting siRNA, the occupancy of GR at the -1007 to -993 GRE was significantly decreased as compared to treatment with non-targeting siRNA, thereby indicating the specificity of GR binding to this GRE (Figure 1). We have reported this new data in Results (subsection “GR Controls PPARα Expression in the Late Fetus”, last paragraph).

5) The paper is written in a complicated fashion, which makes understanding of the most important points tedious. The authors should be encouraged to simplify their writing.

We thank the reviewer for the suggestion. We have reconstructed the manuscript to streamline the story and simplified our writing to enhance accessibility to a broader readership. We believe that the revised manuscript has now been improved to achieve these aims.

Reviewer 3:

1) The mice used are classical PPARα k.o. mice. How specific are the effects foundfor the liver (in particular in vivo) as the PPARα is expressed in many organs(including the gut) and this could influence the outcome?

We agree with the reviewer on this point. PPARα is expressed in many organs including the intestine which is highly challenged in the perinatal life. To further support our findings, we have performed Oil-Red-O staining and histological examination on the livers of hepatocyte-specific *Ppara^-/-^* mice (*Ppara^fl/fl^Alb^Cre/+^*) and their wild-type littermates (*Ppara^fl/fl^*) (Figure 4—figure supplement 1). We demonstrated that *Pparα^hep^*^-/-^ mice exhibited liver steatosis at P3. Hence, we conclude that the phenotypes observed in our model is most likely due to the effects specifically observed in the hepatocytes. This information has been included in the Results (subsection “PPARα Is Required for Protection against Neonatal Hepatic Steatosis”, second paragraph)

*2) The hepatic steatosis in Ppara^-/-^is caused by? The dietary fat or more related to the carb part of the diet leading to de novo lipogenesis that is also under control of PPARα, I thought (references are not mentioned describing the role in regulating 'new' fat metabolism). Translated to the human situation early breastmilk has a rather low fat content and as a baby breastfeeds, the fat content very gradually increases, with the milk becoming fattier and fattier over time. So in the beginning is largely a sweet high lactose/carb feed; how does this relate to this study?*

We thank the reviewer for highlighting these important points. In our opinion, it is the deficiency in metabolising the dietary fat that contributes to this steatotic phenotype in *Ppara^-/-^* livers because in the early days of life mouse milk is abundant in lipids and insulin level is low while glucagon is high, which promotes FAO rather than de novo synthesis of FA (Girard et al., 1977) (doi: 10.1007/978-1-4612-6366-1_36). As discovered by Semenkovich and co-workers (Chakravarthy et al., 2005, Chakravarthy et al., 2009) (PMID:16054078, 19646743), new FA synthesized by fatty acid synthase regulates glucose, lipid, and cholesterol metabolism by serving as endogenous activators of distinct physiological pools of PPARα in adult liver. However, we doubt that de novo synthesis of FA plays a predominant role in the regulation of PPARα activity during early postnatal period since insulin/glucagon ratio is low.

We agree with the reviewer that early human breast milk has a rather relatively low fat content (3%-5%) and a relatively high content of lactose/carbohydrate (6.9%-7.2%). As described by Jenness (1979) (PMID: 392766), fat content does not vary much during lactation but exhibits large diurnal variations and increases during the course of each nursing. Therefore, the human breast milk may be classified as not ketogenic. However, both newborn humans and mouse pups are in ketogenic conditions, in which similar plasma concentrations of ketone body were detected in both species (0.2-0.5 mmol/L) (Hamosh, 2004) (doi:10.1016/B978-0-7216-9654-6.50032-1). This neonatal ketogenic condition could be due to a low plasma insulin/glucagon ratio at this stage of life. Furthermore, many scientists are more inclined towards the notion that lactose and oligosaccharides−two of the main carbohydrates present in the milk, are not used as general fuel, but preferentially to support the growth of the brain and myelin synthesis as well as to provide nutrients for the newly established gut microbiota of infants (Edmond, 1992; Chichlowski et al., 2011) (PMID: 1295662, 22129386). Indeed, many believe that the reason for such a ketogenic environment in human infants is crucial to preserve carbohydrates for the development of the brain (Klepper et al., 2002) (PMID: 12555938). Therefore, oxidative metabolism of lipids represents a preferred fuel for the newborns. We have tried to clarify and discussed this important information in the Discussion (sixth paragraph).

*3) Related to this the HF diets used for the challenge are quite extreme, ketogenic diets, why this is needed as I guess this is not the same in the human situation? I do not see the full relevance for this challenge? To induce FGF21? Is FGF21 functional for early nutrition related metabolic regulation or more related to starvation/low calorie periods of the baby (where β-hydroxybutyrate also goes up?), so is it ketogenesis because of ketogenic diets or fasting-related metabolic states? I am not really convinced of the diets used here to mimic (human) milk. Human milk is clearly not a ketogenic diet. So I question the regulation of FGF21 because of the milk.*

We thank the reviewer for bringing to our attention this lack of clarity. Relating to the previous comment, the purpose of the high-fat diet (HFD) challenge we performed in this experiment was to test whether the change in food composition that occurs at weaning is responsible for the reduction in the fatty liver phenotype as observed in *Ppara^-/-^* pups after P5-6 (Figure 4—figure supplement 1). Indeed, many other changes occur at weaning, including changes in the insulin/glucagon ratio (Girard et al., 1977). The high-fat diet (HFD) experiment indicates that the lipid content of the milk primarily contributes to the development of liver steatosis in *Ppara^-/-^* pups. Our findings suggest that the concomitant presence of mitochondrial dysfunction (cum defective oxidative capacity) and dietary fat synergistically contribute to the development of liver steatosis in *Ppara^-/-^* neonates. As discussed in our response to comment #2, both newborn humans and mouse pups are in ketogenic condition (Medina and Tabernero, 2005) (PMID: 15573408). We believe that the high circulating levels of ketone bodies (e.g., β-hydroxybutyrate) in neonates are due to the use of lipids as the preferred fuel to support the energy demand for growth. This in turn induces FGF21 through HDAC3 inhibition by ketone bodies. We believe that the induction of FGF21 in neonates after milk suckling relates more to metabolic regulation rather than a response to starvation/low calorie condition which should no longer exist upon milk ingestion. This notion is substantiated by the stimulatory effects of FGF21 on hepatic FAO and energy expenditure to prevent hepatic steatosis following milk ingestion (Xu et al., 2009) (PMID: PMC2606881). We have clarified this point in the Results (subsection “PPARα Is Required for Protection against Neonatal Hepatic Steatosis”, second paragraph). We have also discussed this important point in the Discussion (sixth paragraph).

*4) The composition is not precisely mentioned e.g. no lactose or sucrose in the diet?). On the other hand, using chow is not ideal as it will prevent the exact reproducibility of this study because of the natural/variable ingredients of the chow.*

We have provided the formulation for all the diets used in this study (Table 1, Table 4, Table 5). We used an internationally recognised formulation, AIN93M, which specifies the total energy derived from carbohydrate, fat, and protein to be 76%, 9%, and 15%, respectively. Although the ingredients used for each chow diet may vary to suit locally available raw materials, the formulation and the% total energy guideline should allow reproducibility of this study.

*In summary, I am not so convinced yet that the main finding of this paper (epigenetic regulation of FGF21, which has been partly described before, and partly attributed to SIRT1/3) is really related to the HF diet challenge but it could be also alternatively explained as it might contribute in the difficult periods of low-/no-food /starvation when bHB is providing fuel for the brain to optimize glucose transport by other organs to prevent hypoglycemia etc.?*

We thank the reviewer for highlighting this important point. We have discussed the work by Marrero and co-workers (Vilà-Brau et al., 2011) (PMID: 2150232) in which they show that FGF21 is specifically induced by HMGCS2 activity and that the oxidized form of ketone bodies (acetoacetate) could also induce FGF21, possibly via a SIRT1-dependent mechanism.However, as discussed in comments #2 and #3, the induction of FGF21 is likely linked to the ketogenic condition in neonates. It is unclear why neonates exhibit higher circulating levels of ketone bodies. We believe that it could be associated with the use of lipids as preferred fuel for growth while the carbohydrate parts of milk contribute to brain growth and gut microbiota nourishment. Further, the ketogenic condition may also be explained by the low insulin/glucagon ratio in early postpartum life. Regardless, it is clear that ketone bodies act as a hit that is important to prime PPARα-dependent expression of *Fgf21* by modulating HDAC3 activity on *Fgf21* promoter. As suggested by the reviewer, we have tried to further discuss and clarify this important information in the Discussion (fifth and sixth paragraphs).